# BoGrape: Bayesian optimization over graphs with shortest-path encoded

**Yilin Xie**[*] **& Shiqiang Zhang**[*]
Department of Computing, Imperial College London, UK
`{yilin.xie22, s.zhang21}@imperial.ac.uk`

**Jixiang Qing**
Department of Computing, Imperial College London, UK
School of Mathematical Sciences, Lancaster University, UK
`j.qing@imperial.ac.uk`

**Ruth Misener**[†] **& Calvin Tsay**[†]
Department of Computing, Imperial College London, UK
`{r.misener, c.tsay}@imperial.ac.uk`

## Abstract

Graph-structured data are central to many scientific and industrial applications where the goal is to optimize expensive black-box objectives defined over graph structures or node configurations—as seen in molecular design, supply chains, and sensor placement. Bayesian optimization offers a principled approach for such settings, but existing methods largely focus on functions defined over nodes of a fixed graph. Moreover, graph optimization is often approached heuristically, and it remains unclear how to systematically incorporate structural constraints into BO. To address these gaps, we build on shortest-path graph kernels to develop a principled framework for acquisition optimization over unseen graph structures and associated node attributes. Through a novel formulation based on mixed-integer programming, we enable global exploration of the combinatorial domain over graph structures and explicit embedding of problem-specific constraints. We demonstrate that our method, BoGrape, is competitive both on general synthetic benchmarks and representative molecular design case studies with application-specific constraints.

## 1 Introduction

Graph-structured data are playing an emerging role across scientific and industrial fields, giving rise to a series of decision-making problems over graph domains, such as graph-based molecular design (Korovina et al., 2020; Mercado et al., 2021; Yang et al., 2024) and neural architecture search (Elsken et al., 2019; White et al., 2023). Broadly speaking, there are two classes of graph optimization problems (Wan et al., 2023): (i) *optimizing over nodes*, with a given (unknown) graph as the search space and a function over nodes as the objective, and (ii) *optimizing over graphs*, with the entire (constrained) graph domain as the search space and a function over graphs as the objective. The latter case, which this work studies, is usually more challenging since the graph structure itself is optimized, resulting in a complicated combinatorial optimization task.

For both aforementioned scenarios, the objective function can be a black-box, and, when expensive to evaluate, discourages gradient- and population-based methods. These characteristics motivate several works to extend Bayesian optimization (BO) (Frazier, 2018; Garnett, 2023) to graph domains (Cui & Yang, 2018; Oh et al., 2019; Wan et al., 2023; Liang et al., 2024) given its potential sample efficiency. BO relies on two main components: a surrogate model, e.g., Gaussian processes (GPs), trained on available data to approximate the underlying function, and an acquisition function used

---

[*]Equal contributions.
[†]Equal contributions. Corresponding authors.

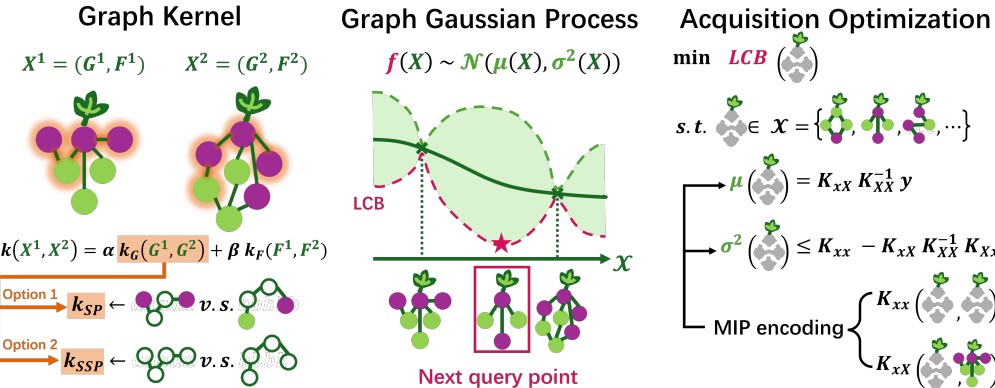

Figure 1: Key components of BoGrape. The **graph kernel** comprises $k_G$ and $k_F$ on the graph and feature levels, resp. The **graph GP** is subsequently trained using the chosen kernel, and its posterior is used to build the acquisition function, e.g., LCB. Note that graph GP includes discrete graph domains; the continuous domain is only for illustration purposes. **Acquisition optimization** is formulated and solved as a MIP using the encoding of shortest paths and graph kernels, giving the next query point.

to suggest the next sample. To translate BO to graph domains, one needs a surrogate model over graph inputs with suitable uncertainty quantification, leading existing approaches to adapt GPs with various graph kernels (Ramachandram et al., 2017; Borovitskiy et al., 2021; Ru et al., 2021; Zhi et al., 2023). However, a general graph BO framework is missing, since existing works either (i) limit the searchable graph set to a given fixed graph (Oh et al., 2019; Wan et al., 2023; Liang et al., 2024), directed labeled graphs (Ru et al., 2021; Wan et al., 2021; White et al., 2021), unlabeled graphs (Cui & Yang, 2018), etc. or (ii) rely on task-specific similarity metrics (Kandasamy et al., 2018).

When optimizing over graphs, the search space includes both continuous and discrete variables, thus limiting the choice of optimization techniques. For example, acquisition function optimization in graph BO is mostly performed using evolutionary algorithms (Kandasamy et al., 2018; Wan et al., 2021) or sampling (Ru et al., 2021; Wan et al., 2023), which are incapable of (i) effectively exploring the search domain, (ii) embedding problem-specific constraints, and (iii) guaranteeing optimality in terms of acquisition function, which is essential for optimization convergence. To mitigate these issues, this paper explores mixed-integer programming (MIP) as an alternative to represent an analytic expression of the graph function. The challenges this paper addresses are to manage both the black-box setting and MIP encodings of surrogates for graph BO.

Recent advances on applying MIP to optimize trained machine learning (ML) models (Ceccon et al., 2022; Schweidtmann et al., 2022; Thebelt et al., 2022b) suggest pathways to address these challenges. By equivalently encoding surrogates, e.g., GPs (Schweidtmann et al., 2021), trees (Mišić, 2020; Mistry et al., 2021; Ammari et al., 2023), neural networks (NNs) (Fischetti & Jo, 2018; Anderson et al., 2020; Tsay et al., 2021; Wang et al., 2023), as constraints in larger decision-making problems, several MIP-based BO methods are proposed, allowing global optimization over mixed-feature domains (Thebelt et al., 2021; 2022a; Papalexopoulos et al., 2022; Xie et al., 2024). Moreover, some works develop MIP-based techniques to handle optimization problems constrained by graph neural networks (GNNs), with applications to molecular design (Zhang et al., 2023; McDonald et al., 2024; Zhang et al., 2024) and robustness certification (Hojny et al., 2024; Gaines et al., 2025). However, given the data requirements of GNNs, the computational cost of solving the large resulting MIPs, and the lack of uncertainty quantification, GNNs are impractical surrogates for graph BO.

This paper proposes BoGrape, a MIP-based graph BO method to optimize functions over connected graphs with attributes. GP surrogates are formulated and optimized using global acquisition function optimization techniques introduced in Xie et al. (2024). We develop four variants of the classic shortest-path graph kernel (Borgwardt & Kriegel, 2005), as well as MIP encodings of graph search space, for use in BoGrape. By introducing a representation of the shortest paths as decision variables, the acquisition function optimization is formulated as a MIP with a mixed-feature search space, graph kernel, and relevant problem-specific constraints. While we choose shortest path kernels for their

ability to model both undirected and directed graphs, BoGrape focuses on undirected graphs. Figure 1 illustrates the BoGrape pipeline. Our main contributions include:

- We propose graph representations with their corresponding shortest paths, and theoretically prove that the feasible domain of our formulation is equivalent to the graph space consisting of all connected graphs.
- We formulate shortest-path graph kernels, node attribute kernels, and GP posterior information based on our graph encoding as MIP constraints, enabling global acquisition optimization.
- We provide a principled BO framework over graph spaces from a discrete optimization viewpoint. BoGrape is compatible with problem-specific constraints over graph structures, node attributes, and their interactions.

## 2 PRELIMINARIES

### 2.1 BAYESIAN OPTIMIZATION (BO)

BO (Frazier, 2018) is a derivative-free optimization framework to iteratively approach the optimum of an expensive-to-evaluate, black-box function. At each iteration, a surrogate model, usually a GP (Schulz et al., 2018), is trained on the current observed dataset. With the surrogate constructed and trained, an acquisition function is then formulated based on the posterior information, e.g., probability of improvement (PI) (Kushner, 1964), expected improvement (EI) (Jones et al., 1998), lower confidence bound (LCB) (Srinivas et al., 2010), predictive Entropy search (PES) (Hernández-Lobato et al., 2014), etc.. Optimizing the acquisition function returns the next query, whose function value is evaluated to form the next data point. This process repeats until meeting a stopping criterion.

### 2.2 GLOBAL OPTIMIZATION OF ACQUISITION FUNCTIONS

Most theoretical results for regret bounds in BO rely on the global optimization over acquisitions (Srinivas et al., 2012), i.e., they assume the global minimizer/maximizer of the acquisition function is found at each step, which may not be satisfied using gradient- and sample-based optimizers. Xie et al. (2024) introduce PK-MIQP, a global acquisition optimization framework based on mixed-integer quadratic programming (MIQP). The core of PK-MIQP is the piecewise linearization of a stationary or dot-product kernel, e.g., RBF, Matérn, etc., based on which the acquisition optimization is then formulated as an MIQP. PK-MIQP is useful because of its (i) compatibility with various kernels (note the piecewise linearization is unnecessary if the kernel can be expressed linearly), and (ii) theoretical guarantee on regret bounds. We present its formulation for the LCB acquisition function here:

$$\min \mu - \beta_t^{1/2}\sigma \qquad\qquad \leftarrow \text{ LCB acquisition} \qquad (1a)$$
$$\text{s.t. } \mu = K_{xX}K_{XX}^{-1}\boldsymbol{y} \qquad\qquad \leftarrow \text{ GP posterior mean} \qquad (1b)$$
$$\sigma^2 \le K_{xx} - K_{xX}K_{XX}^{-1}K_{Xx} \qquad\qquad \leftarrow \text{ GP posterior variance} \qquad (1c)$$
$$K_{xX^i} = k(x, X^i), \ \forall 1 \le i < t \qquad\qquad \leftarrow \text{ kernel function} \qquad (1d)$$
$$x \in \mathcal{X} \qquad\qquad \leftarrow \text{ search space} \qquad (1e)$$

### 2.3 SHORTEST-PATH GRAPH KERNELS

Graph kernels extend the concept of kernels to graph domains and are used to measure the similarity between two graphs. Mathematically, a graph kernel $k(\cdot,\cdot) : \mathcal{G} \times \mathcal{G} \to \mathbb{R}$ is given by $k(G, G') = \langle \phi(G), \phi(G') \rangle_{\mathcal{H}}$, where $\phi : \mathcal{G} \to \mathcal{H}$ is a feature map from graph domain $\mathcal{G}$ to a reproducing kernel Hilbert space $\mathcal{H}$ with inner product $\langle \cdot, \cdot \rangle_{\mathcal{H}}$ (Kriege et al., 2020). Past research develops graph kernels using a variety of graph patterns, e.g., neighborhoods, subgraphs, walks, paths. We refer the reader to (Vishwanathan et al., 2010; Borgwardt et al., 2020; Kriege et al., 2020; Nikolentzos et al., 2021) for more details on graph kernels. Several works also use graph kernels to *optimize over nodes* (Oh et al., 2019; Borovitskiy et al., 2021; Wan et al., 2023; Liang et al., 2024), but the involved kernels measure the similarity of two nodes on *one given* graph and do not support *optimizing over graphs* (see Section 1 for this distinction). We focus on the shortest-path (SP) kernel (Borgwardt & Kriegel,

2005) in this paper, owing to its ability to (i) handle both directed and undirected graphs, (ii) consider node labels, and (iii) capture the relationship between non-adjacent graph nodes, making it more general than kernels based on subgraph patterns (Shervashidze et al., 2009; Costa & Grave, 2010). We further discuss the choice of kernels in Appendix B.1. For graph $G$, denote $l_u$ as the label of node $u$, $e_{u,v}$ as the shortest path from $u$ to $v$ (which may not be unique), and $d_{u,v}$ as the shortest distance from node $u$ to $v$ (which is unique). The SP kernel between graphs $G^1 = (V^1, E^1)$ and $G^2 = (V^2, E^2)$ is defined as:

$$k_{SP}(G^1, G^2) = \sum_{u_1, v_1 \in V^1, u_2, v_2 \in V^2} k_v(l_{u_1}, l_{u_2}) \cdot k_e(d_{u_1,v_1}, d_{u_2,v_2}) \cdot k_v(l_{v_1}, l_{v_2}). \tag{2}$$

where $k_v$ is a kernel comparing node labels and $k_e$ is a kernel comparing path lengths.

## 3 METHODOLOGY

### 3.1 VARIANTS OF THE SHORTEST-PATH GRAPH KERNELS

We build on Eq. (2) and develop variants of the shortest-path kernel. Both $k_v$ and $k_e$ in Eq. (2) are usually chosen as Dirac kernels, giving the explicit representation of the SP kernel as:

$$k_{SP}(G^1, G^2) = \frac{1}{n_1^2 n_2^2} \sum_{u_1, v_1 \in V^1, u_2, v_2 \in V^2} \mathbf{1}(l_{u_1} = l_{u_2}, d_{u_1,v_1} = d_{u_2,v_2}, l_{v_1} = l_{v_2}), \tag{SP}$$

where $n_1^2 n_2^2$ is a normalizing coefficient with $n_1, n_2$ as the node numbers of $G^1, G^2$, resp.

Each node may additionally have problem-specific features beyond a single label. From here on, we use $X = (G, F)$ to denote an attributed graph with $G$ as the underlying labeled graph and $F$ as node features. Intuitively, we can compare the features of two nodes instead of labels in $k_v$. However, this could unnecessarily reduce the number of matching paths between two graphs, as requiring identical node features is restrictive and may introduce additional subgraph information into path comparison. Another option is to use a more complicated kernel $k_v$ that measures similarity between features of two nodes, which may significantly increase the computational cost of optimization (similarity is computed for all node pairings). Therefore, we borrow from (Cui & Yang, 2018) the idea to separate the implicit and explicit information of graphs, i.e., the kernel value between two attributed graphs $X^1, X^2$ becomes:

$$k(X^1, X^2) = \alpha \cdot k_G(G^1, G^2) + \beta \cdot k_F(F^1, F^2), \tag{3}$$

where $k_G$ is any graph kernel, e.g., (SP), $k_F$ is any kernel over features, and $\alpha, \beta$ are learnable parameters controlling the trade-off between graph similarity and feature similarity.

Since node label is usually included as a node feature and considered in $k_F$ term, and comparing labels in Eq. (SP) increases the complexity of our upcoming optimization formulations, we further propose a simplified shortest-path (SSP) kernel corresponding to an unlabeled SP kernel:

$$k_{SSP}(G^1, G^2) = \frac{1}{n_1^2 n_2^2} \sum_{u_1, v_1 \in V^1, u_2, v_2 \in V^2} \mathbf{1}(d_{u_1,v_1} = d_{u_2,v_2}). \tag{SSP}$$

**Lemma 3.1.** *SP and SSP kernels are positive definite (PD).*

*Proof.* Borgwardt & Kriegel (2005) prove the SP kernel is PD. The SSP kernel is a special case of the SP kernel where all nodes have the same label, hence is also PD. □

Observe that both the SP and SSP kernels are linear kernels if we pre-compute all shortest paths in each graph and count the number of occurrence for each shortest path length. Such linearity simplifies the optimization step (which still requires the non-trivial representation of shortest paths), but reduces the representation ability of the kernels and limits the maximal rank of the Gram matrix. Motivated by the practically strong performance of exponential kernels such as RBF, Matérn, graph diffusion kernel (Oh et al., 2019), etc., we propose two nonlinear graph kernels based on SP and SSP kernels:

$$k_{ESP}(G^1, G^2) = \exp(k_{SP}(G^1, G^2))/\sigma_k^2, \tag{ESP}$$

$$k_{ESSP}(G^1, G^2) = \exp(k_{SSP}(G^1, G^2))/\sigma_k^2, \tag{ESSP}$$

where variance $\sigma_k^2$ is added to control the magnitude of kernel value.

Table 1: List of variables introduced to represent the shortest path, where $n$ is the number of nodes.

| variables | type | description |
|---|---|---|
| $A_{u,v} \in \{0,1\},\ u,v \in [n]$ | binary | the existence of edge from node $u$ to $v$ |
| $d_{u,v} \in [n],\ u,v \in [n]$ | integer | the length of shortest path from node $u$ to $v$ |
| $\delta^w_{u,v} \in \{0,1\},\ u,v,w \in [n]$ | binary | the presence of node $w$ on the shortest path from $u$ to $v$ |

**Lemma 3.2.** *ESP and ESSP kernels are PD.*

*Proof.* SP and SSP kernels can be rewritten into linear forms, so ESP and ESSP are exponential kernels, which are known to be PD (Fukumizu, 2010). □

*Remark* 3.3. The nonlinear kernels introduce additional difficulties for optimization, as discussed later in Section 3.4, but may demonstrate better empirical performance compared to their linear counterparts, owing to increased representation ability.

## 3.2 GLOBAL ACQUISITION FUNCTION OPTIMIZATION

To extend the prior formulation in Eq. (1) to optimize LCB acquisition in graph space (see Appendix B.2 for the applicability to other acquisition functions), we replace Eq. (1d) by our graph kernels in Section 3.1 and define a combinatorial graph search space for Eq. (1e) as $x = (G, F) \in \mathcal{X} = \mathcal{G} \times \mathcal{F}$. To maintain consistency with the general BO setting, we denote $x = (G, F)$ as the next sample and $X = \{(G^i, F^i), y^i\}_{i=1}^{t-1}$ as the previous samples at the $t$-th iteration. The difference is that now we need to optimize over both the graph domain $G \in \mathcal{G}$ and the feature domain $F \in \mathcal{F}$. W.l.o.g., assume that each node has $M$ features $F^i \in \mathbb{R}^{n(G^i) \times M}$, and the first $L$ features denote the one-hot encoding of its label, i.e., $\sum_{l \in [L]} F^i_l = 1$, where $[n]$ denotes set $\{0, 1, \ldots, n-1\}$. This modified formulation allows: (i) discrete variables, which is a key challenge of graph optimization, (ii) problem-specific constraints over graph domain, and (iii) theoretical guarantees on regret bounds.

A binary adjacency matrix is sufficient to represent the graph domain; however, encoding corresponding shortest-path information (for an unknown graph) is not straightforward and comprises a main technical contribution of this work. We first introduce the formulation of shortest paths in Section 3.3 and then explicitly derive Eq. (1d) in Section 3.4 for the graph kernels in Section 3.1.

## 3.3 ENCODING OF THE SHORTEST PATHS AS OPTIMIZATION CONSTRAINTS

For the sake of exposition, we first consider all connected graphs $G$ with fixed size, i.e., node number $n$ is given (Appendix A.3 discusses formulations for graphs of unknown size). Table 1 summarizes the optimization variables. Since our formulations involve constant graph information and their variable counterparts, for each variable $Var$, we use $Var(G)$ to denote its value on a given graph $G$. For example, $d_{u,v}(G)$ is the shortest distance from node $u$ to node $v$ in graph $G$.

If graph $G$ is given, all variables in Table 1 can be computed using classic shortest-path algorithms, such as the Floyd–Warshall algorithm (Floyd, 1962). In graph optimization tasks, however, we need to encode the relationships between these variables as constraints. Motivated by the Floyd–Warshall algorithm, we first present the constraints in Eq. (5) of Appendix A.2 and then prove there exists a bijective between the feasible domain given by these constraints and all connected graphs with size $n$. Here we directly give the final encoding of the shortest paths in the following linear MIP (details in

Appendix A.2):

$$\begin{cases} A_{v,v} = 1, \ d_{v,v} = 0, \ \delta_{v,v}^w = \mathbf{1}(w = v) & \forall v, w \in [n] \\ d_{u,v} \le 1 + n \cdot (1 - A_{u,v}), & \forall u, v \in [n], \ u \ne v \\ d_{u,v} \ge 2 - A_{u,v}, & \forall u, v \in [n], \ u \ne v \\ d_{u,v} \le d_{u,w} + d_{w,v} - (1 - \delta_{u,v}^w), & \forall u, v, w \in [n] \\ d_{u,v} \ge d_{u,w} + d_{w,v} - 2n \cdot (1 - \delta_{u,v}^w), & \forall u, v, w \in [n] \\ \delta_{u,v}^u = \delta_{u,v}^v = 1, & \forall u, v \in [n], \ u \ne v \\ \sum_{w \in [n]} \delta_{u,v}^w \le 2 + (n - 2) \cdot (1 - A_{u,v}), & \forall u, v \in [n], \ u \ne v \\ \sum_{w \in [n]} \delta_{u,v}^w \ge 2 + (1 - A_{u,v}), & \forall u, v \in [n], \ u \ne v \end{cases} \quad \text{(MIP-SP)}$$

**Lemma 3.4.** $(A_{u,v}(G), d_{u,v}(G), \delta_{u,v}^w(G))$ *is a feasible solution of Eq. (MIP-SP) with size* $n = n(G)$ *given any connected graph* $G$.

*Proof.* Trivial to verify by definition. $\qquad\qquad\qquad\qquad\qquad\qquad\qquad\qquad\qquad\qquad\square$

**Theorem 3.5.** *Given any* $n \in \mathbb{Z}^+$, *for any feasible solution* $(A_{u,v}, d_{u,v}, \delta_{u,v}^w)$ *of Eq. (MIP-SP) with size* $n$, *there exists a unique graph* $G$ *such that:*

$$(A_{u,v}(G), d_{u,v}(G), \delta_{u,v}^w(G)) = (A_{u,v}, d_{u,v}, \delta_{u,v}^w),$$

*i.e., there is a bijection between the feasible domain of Eq. (MIP-SP) with size* $n$ *and the set consisting of all connected graphs with* $n$ *nodes.*

The formulation becomes more complicated when the graph size is unknown (but bounded). Denote $n_0$ and $n$ as the minimal and maximal node numbers, resp., and use $A_{v,v}$ to represent the existence of node $v$. Variables $d_{u,v}$ and $\delta_{u,v}^w$ need to be properly assigned when either $u$ or $v$ does not exist. Moreover, we extend the domain of $d_{u,v}$ from $[n]$ to $[n + 1]$ and use $n$ to denote infinity. Eq. (MIP-SP-plus) in Appendix A.3 presents the encoding and Theorem 3.6 extends Theorem 3.5 to unknown size.

**Theorem 3.6.** *There is a bijection between the feasible domain of Eq. (MIP-SP-plus) with size* $[n_0, n]$ *and all connected graphs with number of nodes in* $[n_0, n]$.

See Appendix A.4 for proofs of Theorems 3.5 and 3.6, which guarantee the equivalence of our encoding for directed and strong connected graphs. Appendix A.6 shows how to further simplify our encoding for undirected graphs. Appendix B.3 discusses the effectiveness of our encoding.

## 3.4 ENCODING OF GRAPH KERNELS AS OPTIMIZATION CONSTRAINTS

We now rewrite Eq. (1d) using Eq. (3) as:

$$K_{xX_i} = k(x, X_i) = \alpha \cdot k_G(G, G^i) + \beta \cdot k_F(F, F^i).$$

Given that $k_F$ is independent of the choice of graph kernel $k_G$, and that kernels on continuous features are studied in Xie et al. (2024), here we focus on formulating $k_G$. See Appendix A.5 for respective kernel encoding with binary features.

Formulating $k_G(G, G^i)$ is straightforward for SP and SSP kernels:

$$k_{SSP}(G, G^i) = \frac{1}{n^2 n_i^2} \sum_{u_1, v_1 \in [n]} \sum_{u_2, v_2 \in [n(G^i)]} d_{u_1, v_1}^{d_{u_2, v_2}(G^i)} = \frac{1}{n^2 n_i^2} \sum_{u,v,s \in [n]} D_s(G^i) \cdot d_{u,v}^s,$$

where $n_i := n(G^i)$ is the node number of $G^i$, $d_{u,v}^s = \mathbf{1}(d_{u,v} = s)$ are indicator variables:

$$\sum_{s \in [n+1]} d_{u,v}^s = 1, \quad \sum_{s \in [n+1]} s \cdot d_{u,v}^s = d_{u,v}, \ \forall u, v \in [n],$$

and $D_s(G^i)$ is the number of shortest paths with length $s$ in $G^i$:

$$D_s(G^i) = |\{(u, v) \mid u, v \in [n_i], \ d_{u,v}(G^i) = s\}|.$$

---

**Algorithm 1** BoGrape at $t$-th iteration.

---

1: **Input:** dataset $X = \{(G^i, F^i), y^i\}_{i=1}^{t-1}$, hyperparameter $\beta_t$, graph kernel
2: **Model training:** kernel parameters $\alpha, \beta, \sigma_k^2$          ▷ graph GP fit to $X$
3: **Acquisition formulation:**
4:     represent $K_{xX^i}$ and $K_{xx}$ in Eqs. (1b) – (1d)          ▷ Section 3.4
5:     search space $\mathcal{X}$ in Eq. (1e)          ▷ problem-specific
6: **Optimization:** initialize and solve MIP Eq. (1)          ▷ global optimization
7: **Output:** proposed sample $(G^t, F^t)$

---

*Remark* 3.7. $d_{u,v}^n$ is not used in evaluating the kernel, since it means the shortest path does not exist.

Similarly, introducing indicator variables $p_{u,v}^{s,l_1,l_2}$ as:

$$p_{u,v}^{s,l_1,l_2} = \mathbf{1}(F_{u,l_1} = 1, \ d_{u,v} = s, \ F_{v,l_2} = 1), \ \forall u, v, s \in [n], \ l_1, l_2 \in [L],$$

and counting the numbers of each type of paths in $G^i$:

$$P_{s,l_1,l_2}(G^i) = |\{(u,v) \mid u, v \in [n_i], \ l_u(G^i) = l_1, \ d_{u,v}(G^i) = s, \ l_v(G^i) = l_2\}|,$$

the SP kernel is formulated as:

$$k_{SP}(G, G^i) = \frac{1}{n^2 n_i^2} \sum_{u,v,s \in [n], l_1, l_2 \in [L]} P_{s,l_1,l_2}(G^i) \cdot p_{u,v}^{s,l_1,l_2}.$$

There are several ways to handle the exponential kernels: (i) directly use (local) nonlinear solvers, losing optimality guarantees, (ii) piecewise linearize the exponential function following Xie et al. (2024), or (iii) utilize nonlinear MIP functionalities in established solvers such as Gurobi (Gurobi Optimization, LLC, 2024) or SCIP (Vigerske & Gleixner, 2018). In our experiments, we choose to use Gurobi, which by default employs a dynamic outer approximation of the exponential function given an error tolerance.

It is noteworthy that $K_{xx}$ in Eq. (1c) is not constant with a non-stationary kernel, making it the most complicated term in the whole formulation. By definition, $k_{SSP}(G, G)$ has a quadratic form:

$$k_{SSP}(G, G) = \frac{1}{n^4} \sum_{s \in [n]} D_s^2,$$

where $D_s = \sum_{u,v \in [n]} d_{u,v}^s, \ \forall s \in [n]$. Reusing the indicator trick and introducing $D_s^c = \mathbf{1}(D_s = c)$, the quadratic form is equivalently linearized as:

$$K_{SSP}(G, G) = \frac{1}{n^4} \sum_{s \in [n], c \in [n^2+1]} c^2 \cdot D_s^c,$$

where indicator variables $D_s^c, \ \forall s \in [n], c \in [n^2 + 1]$ should satisfy:

$$\sum_{c \in [n^2+1]} D_s^c = 1, \quad \sum_{c \in [n^2+1]} c \cdot D_s^c = D_s, \ \forall s \in [n].$$

Repeating the procedure for the SP kernel, we have:

$$K_{SP}(G, G) = \frac{1}{n^4} \sum_{s \in [n], l_1, l_2 \in [L], c \in [n^2+1]} c^2 \cdot P_{s,l_1,l_2}^c,$$

where indicator variables $P_{s,l_1,l_2}^c = \mathbf{1}(P_{s,l_1,l_2} = c), \ \forall s \in [n], \ l_1, l_2 \in [L], \ c \in [n^2 + 1]$ satisfy:

$$\sum_{c \in [n^2+1]} P_{s,l_1,l_2}^c = 1, \quad \sum_{c \in [n^2+1]} c \cdot P_{s,l_1,l_2}^c = P_{s,l_1,l_2}, \ \forall s \in [n], \ l_1, l_2 \in [L].$$

With the above graph GP model and optimization encodings, we have presented all the pieces needed to implement an end-to-end graph BO procedure. Algorithm 1 outlines BoGrape.

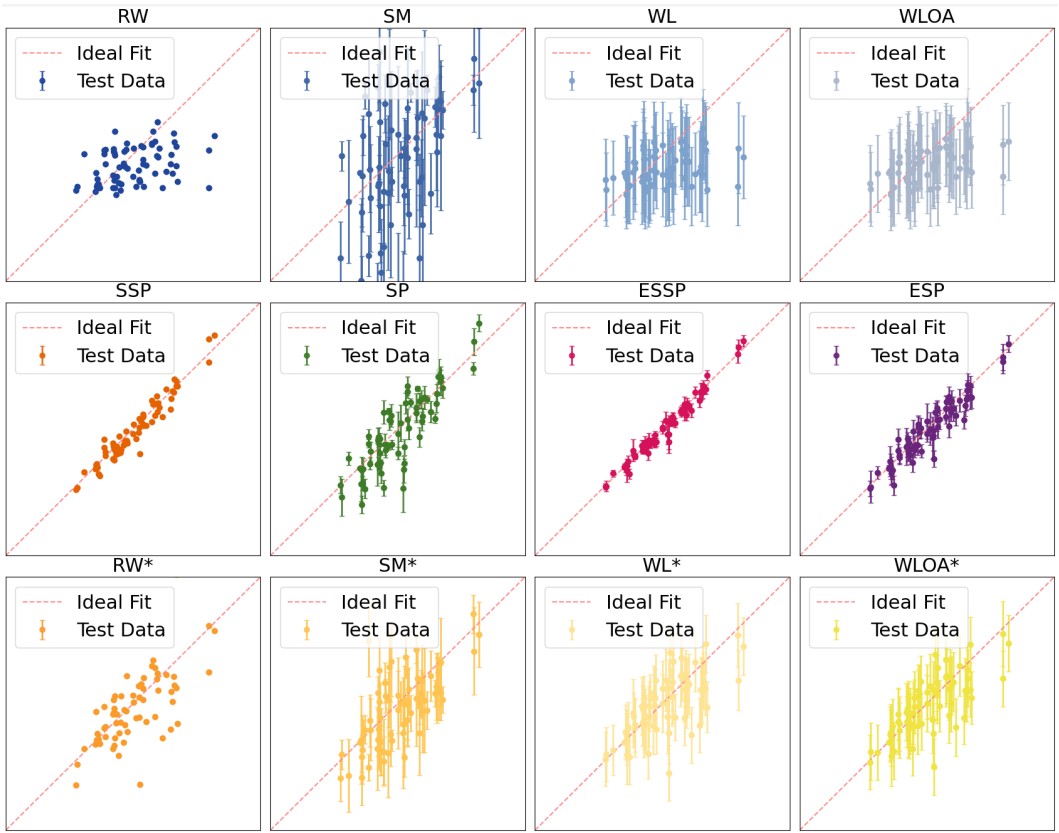

Figure 2: Compare predictive performance of GP with different kernels. * indicates linear combination of given kernel and feature kernel. 100 samples are randomly chosen from the QM7 dataset with various graph sizes, 30 of which are used for training. The predictive mean with one standard deviation (predicted $y$) of the remaining 70 graphs are plotted against their real values (true $y$).

## 4 EXPERIMENTS

All experiments are performed on a 4.2 GHz Intel Core i7-7700K CPU with 16 GB memory. We use GPflow (Matthews et al., 2017) to implement GP models, GraKel (Siglidis et al., 2020) to implement the classic graph kernels, PyG (Fey & Lenssen, 2019) to implement GNNs, and Gurobi (Gurobi Optimization, LLC, 2024) to solve MIPs.

There are few synthetic benchmark functions $f : \mathcal{G} \times \mathcal{F} \rightarrow \mathbb{R}$ for general graph domains since most graph BO works focus on specific types of graphs. W,o.l.g., we consider GNNs as graph functions that maps general labeled connected graph to real values. We conduct experiments considering two settings: (i) randomly initialized GNNs which serves as random synthetic functions, and (ii) GNNs trained on molecular datasets as graph property predictor in real-world case studies. It is noteworthy that the architectures of GNNs, the training mechanism, and the choice of datasets are not major components of our work, since they merely serve as benchmarks of black-box graph functions.

### 4.1 MODEL PERFORMANCE

Before conducting the optimization tasks, we first compare the performance of graph GPs with various graph kernels on randomly sampled molecules from the QM7 dataset (Blum & Reymond, 2009; Rupp et al., 2012). Figure 2 visualizes the results and more detailed discussion can be found in Appendix C.1. Figure 2 shows that four shortest-path kernels have comparable prediction accuracy against other classic graph kernels, while the two exponential kernels quantify uncertainty more accurately (also supported by Table 4 of Appendix C.1). Additionally, the introduction of the additional feature kernel term generally improves the predictive performance of all kernels. For larger graph sizes, Table 3

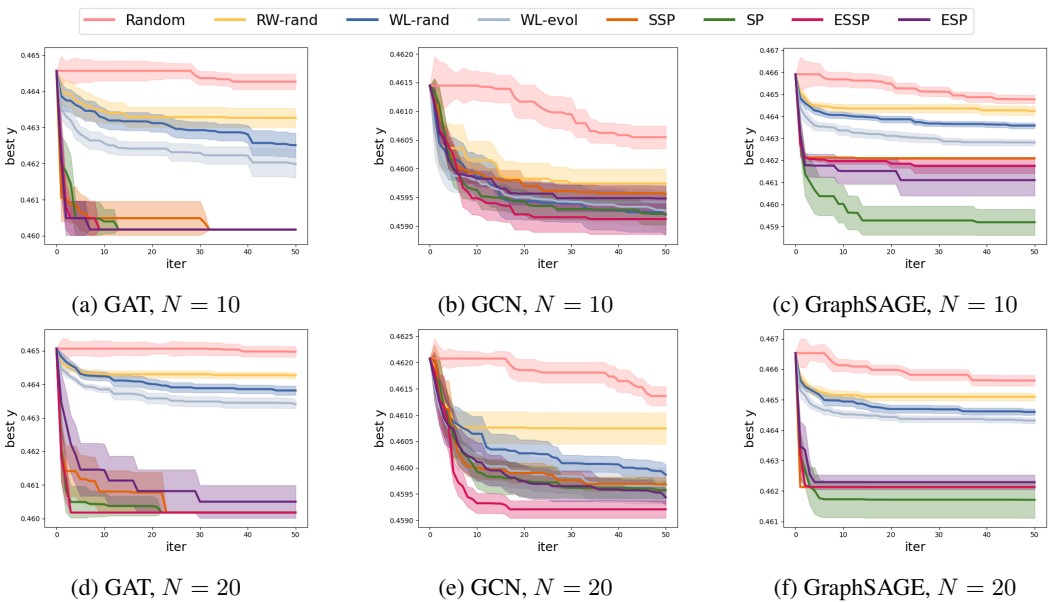

(a) GAT, $N = 10$  (b) GCN, $N = 10$  (c) GraphSAGE, $N = 10$

(d) GAT, $N = 20$  (e) GCN, $N = 20$  (f) GraphSAGE, $N = 20$

Figure 3: Bayesian optimization results on synthetic benchmarks with $N \in \{10, 20\}$. Best objective value is plotted at each iteration. Mean with 0.5 standard deviation over 10 replications is reported.

of Appendix C.1 shows that the more complicated kernels, i.e., SP and ESP, are generally better at predicting graph properties, since they impose stronger criteria on comparing shortest-paths between two graphs.

*Remark* 4.1. See Appendix C.1 for similar illustrations of other graph kernels and more evaluations, and Appendix B.4 for complexity analysis of different graph kernels.

## 4.2 Optimization of synthetic benchmarks

We first evaluate the performance of BoGrape over synthetic benchmarks, i.e., arbitrary functions over graphs. Specifically, given their ability as universal approximators, we cover the range of black-box graph functions using randomly initialized GNNs including GAT (Veličković et al., 2017), GCN (Kipf & Welling, 2017), and GraphSAGE (Hamilton et al., 2017). Each GNN consists of two convolutional layers to learn graph embeddings, and two linear layers. Each hidden layer has 64 features. The search space is the set of all connected, undirected graphs with $N$ nodes, and each node has one-hot features with length $L = 5$ as its label. We propose these functions as benchmark problems, given: (i) there are no existing synthetic benchmarks in graph BO literature, (ii) these benchmarks impose neither problem-specific constraints nor assumptions over the graph space (except for connectivity), making them suitable for comparison of a wide class of methods. These benchmark functions for graph BO are available at: [link to be added after peer review].

BoGrape is compared against the following baselines: (i) Random: random sampling, i.e., randomly sample one connected graph at each iteration, (ii) RW-rand: use graph GP with random walk (RW) kernel as surrogate, and sampling-based acquisition optimization, i.e., choosing the sample with the best LCB value among 20 random graphs, (iii) WL-rand: use Weisfeiler-Lehma (WL) kernel in RW-rand, (iv) WL-evol: use evolutionary algorithm for acquisition optimization in WL-rand.

*Remark* 4.2. WL-rand and WL-evol are adapted from Ru et al. (2021), which is specifically designed for neural architecture search (NAS). WL-evol could be regarded as the state-of-the-art method in graph BO.

For each benchmark with size $N = 10, 20$, we conduct BO with 10 initial samples and 50 iterations. When solving Eq. (1), we observed good solutions to be found early (since more time is spent on proving optimality) and set 600s as the MIP time limit. As shown in Figure 3, BoGrape with all kernel variants outperforms baselines in most cases. When the graph size is small, SP and ESP perform better, since they are more expressive. For larger sizes, using simpler kernels reduces model

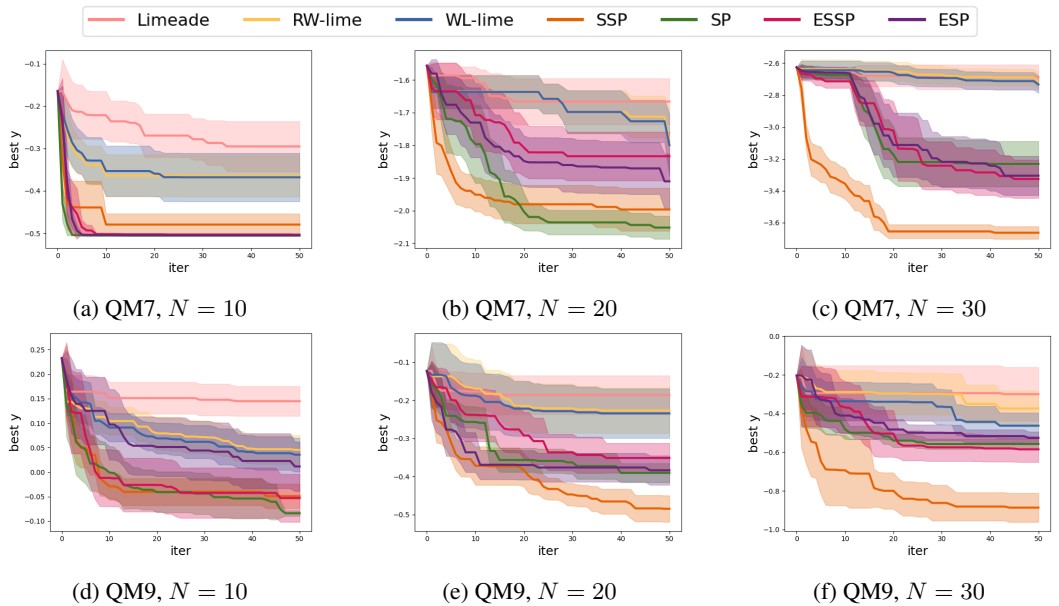

Figure 4: Bayesian optimization results on QM7 and QM9 with $N \in \{10, 20, 30\}$. Best objective value is plotted at each iteration. Mean with 0.5 standard deviation over 10 replications is reported.

complexity and produces better solutions within the given time limit. The trade-off is between the expressiveness of kernel and the complexity of the resulting optimization problem. We further discuss this computational limitation in Appendix B.5 and possible solutions in Appendix B.6 and kernel selection in Appendix B.8.

## 4.3 REAL-WORLD CASE STUDY

We next consider optimal molecular design (McDonald et al., 2024; Zhang et al., 2024) as a real-world case study. Following Zhang et al. (2023), we train two GNNs on dataset QM7 (Blum & Reymond, 2009; Rupp et al., 2012) and QM9 (Ruddigkeit et al., 2012; Ramakrishnan et al., 2014) as oracle predictors, i.e., the functions that we seek to optimize. The additional challenge of this task compared to Section 4.2 is that molecules are not arbitrary labeled graphs: the molecular graph should be compatible with atom features. We found these structural constraints to effectively prevent the sampling- and evolutionary-based methods used in Section 4.2 from producing feasible solutions. Therefore, we modify Random to only consider randomly generated feasible molecules from Limeade (Zhang et al., 2025), and we remove WL-evol from comparison. We also adapt molecular feasibility constraints from Limeade and add them to our MIP formulation to ensure only valid molecules are considered during the optimization. Further related details are given in Appendix C.3. The computational results in Figure 4 show that BoGrape again generally outperforms baselines regardless of which kernel is used. We discuss this application further in Appendix B.7.

## 5 CONCLUSION

This work proposes BoGrape to optimize black-box functions over graphs. Four shortest-path graph kernels are presented and tested on both prediction and Bayesian optimization tasks. The underlying mixed-integer formulation provides a flexible and general platform including mixed-feature search spaces, graph kernels, acquisition functions, and problem-specific constraints. Our results show promising performance and suggest trade-offs between query-efficiency and computational time when choosing a suitable kernel. Future work may further simplify the formulations of BoGrape and relax the requirement on graph connectivity.

ACKNOWLEDGMENTS

The authors gratefully acknowledge support from a Department of Computing Scholarship (YX), BASF SE, Ludwigshafen am Rhein (SZ), Engineering and Physical Sciences Research Council [grant numbers EP/W003317/1 and EP/X025292/1] (RM, CT, JQ), Research England's Expanding Excellence in England (E3) fund awarded to MARS (Mathematics for AI in Real-world Systems) at Lancaster University (JQ), a BASF/RAEng Research Chair in Data-Driven Optimisation (RM), a BASF/RAEng Senior Research Fellowship (CT).

REPRODUCIBILITY STATEMENT

We take the following measures to facilitate the reproducibility of our work. Theoretical contributions are detailed and explained in both Section 3 in the main paper and Appendix A. Theoretical claims are supported by formal proofs provided in Appendix A.4. To support the replication of empirical findings, we provide code implementations including our method, synthetic graph functions mentioned in Section 4.2 and models used in experiments in the supplementary materials.

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

## A    ENCODING OF GRAPH KERNELS

### A.1    NOTATIONS

We provide details for all variables introduced in this paper in Table 2. Recall that the search domain considered here consists of all connected graphs with node number ranging from $n_0$ to $n$, each node has $M$ binary features with the first $L$ node features as the one-hot encoding of node label.

Table 2: All variables introduced in the optimization formulation for graph kernels.

| Variables | Domain | Description |
|---|---|---|
| $A_{u,v}$, $u,v \in [n]$ | $\{0,1\}$ | the existence of edge from $u$ to $v$ |
| $d_{u,v}$, $u,v \in [n]$ | $[n+1]$ | the length of shortest path from $u$ to $v$ |
| $\delta_{u,v}^w$, $u,v,w \in [n]$ | $\{0,1\}$ | if $w$ appears at the shortest path from $u$ to $v$ |
| $d_{u,v}^s$, $u,v \in [n], s \in [n+1]$ | $\{0,1\}$ | indicator: $\mathbf{1}(d_{u,v}=s)$ |
| $D_s$, $s \in [n]$ | $[n^2+1]$ | # shortest paths with length $s$ |
| $D_s^c$, $s \in [n], c \in [n^2+1]$ | $\{0,1\}$ | indicator: $\mathbf{1}(D_s=c)$ |
| $p_{u,v}^{s,l_1,l_2}$, $u,v,s \in [n], l_1,l_2 \in [L]$ | $\{0,1\}$ | indicator: $\mathbf{1}(F_{u,l_1}=1, d_{u,v}=s, F_{v,l_2}=1)$ |
| $P_{s,l_1,l_2}$, $s \in [n], l_1,l_2 \in [L]$ | $[n^2+1]$ | # shortest paths with length $s$ and labels $l_1, l_2$ |
| $P_{s,l_1,l_2}^c$, $s \in [n], l_1,l_2 \in [L], c \in [n^2+1]$ | $\{0,1\}$ | indicator: $\mathbf{1}(P_{s,l_1,l_2}=c)$ |
| $N^m$, $m \in [M]$ | $[N+1]$ | sum of $m$-th feature over all nodes |
| $N_m^c$, $m \in [M], c \in [M+1]$ | $\{0,1\}$ | indicator: $\mathbf{1}(N_m=c)$ |

### A.2    SHORTEST PATH ENCODING FOR GRAPHS WITH FIXED SIZE

We first present necessary conditions that $A_{u,v}, d_{u,v}, \delta_{u,v}^w$ should satisfy in Eq. (5):

$$A_{v,v} = 1, \qquad\qquad\qquad\qquad \forall v \in [n] \tag{5a}$$

$$d_{v,v} = 0, \qquad\qquad\qquad\qquad \forall v \in [n] \tag{5b}$$

$$d_{u,v} \begin{cases} =1, & A_{u,v}=1 \\ >1, & A_{u,v}=0 \end{cases}, \qquad \forall u,v \in [n],\ u \neq v \tag{5c}$$

$$d_{u,v} \begin{cases} =d_{u,w}+d_{w,v}, & \delta_{u,v}^w=1 \\ <d_{u,w}+d_{w,v}, & \delta_{u,v}^w=0 \end{cases}, \qquad \forall u,v \in [n],\ u \neq v \tag{5d}$$

$$\delta_{v,v}^w = \begin{cases} 1, & w=v \\ 0, & w \neq v \end{cases}, \qquad \forall v \in [n] \tag{5e}$$

$$\delta_{u,v}^u = \delta_{u,v}^v = 1, \qquad\qquad \forall u,v \in [n],\ u \neq v \tag{5f}$$

$$\sum_{w \in [n]} \delta_{u,v}^w \begin{cases} =2, & A_{u,v}=1 \\ >2, & A_{u,v}=0 \end{cases}, \qquad \forall u,v \in [n],\ u \neq v \tag{5g}$$

Eq. (5) restricts $A_{u,v}, d_{u,v}, \delta_{u,v}^w$ in the following rules:

- Eq. (5a) initializes the diagonal elements.

- Eq. (5b) initializes the shortest distance from $v$ to itself.

- Eq. (5c) forces the shortest distance from node $u$ and $v$ be 1 if edge $u \to v$ exists, and larger than 1 otherwise.
  Rewrite Eq. (5c) as:

$$d_{u,v} \leq 1 + n \cdot (1 - A_{u,v}), \quad \forall u,v \in n],\ u \neq v$$
$$d_{u,v} \geq 2 - A_{u,v}, \qquad\qquad \forall u,v \in [n],\ u \neq v$$

  where $n$ is a big-M coefficient using $d_{u,v} \leq n-1$.

- Eq. (5d) is the triangle inequality for distance matrix $d$.
  Rewrite Eq. (5d) as:

$$d_{u,v} \leq d_{u,w} + d_{w,v} - (1 - \delta_{u,v}^w), \qquad \forall u, v, w \in [n]$$
$$d_{u,v} \geq d_{u,w} + d_{w,v} - 2n \cdot (1 - \delta_{u,v}^w), \quad \forall u, v, w \in [n]$$

where $2n$ is a big-M coefficient since $d_{u,w} + d_{w,v} < 2n$.
- Eq. (5e) initializes $\delta_{v,v}^w$ by definition.
- Eq. (5f) initializes $\delta_{u,v}^u$ and $\delta_{u,v}^v$ by definition.
- Eq. (5g) ensures that there is at least one node at the shortest path from node $u$ to $v$ if there is no edge from node $u$ to $v$. Otherwise, no node except for $u$ and $v$ could appear at the shortest path from $u$ to $v$.
  Rewrite Eq. (5g) as:

$$\sum_{w \in [n]} \delta_{u,v}^w \leq 2 + (n - 2) \cdot (1 - A_{u,v}), \quad \forall u, v \in [n], \ u \neq v$$
$$\sum_{w \in [n]} \delta_{u,v}^w \geq 2 + (1 - A_{u,v}), \qquad \forall u, v \in [n], \ u \neq v$$

where $n - 2$ is a big-M coefficient since $\sum_{w \in [n]} \delta_{u,v}^w \leq n$.

Replacing disjunctive constraints accordingly in Eq. (5) gives the final formulation Eq. (MIP-SP).

### A.3 SHORTEST PATH ENCODING FOR GRAPH WITH UNKNOWN SIZE

We extend constraints in Eq. (5) to handle changeable graph size. Full constraints are as follows:

$$A_{v,v} \geq A_{v+1,v+1}, \qquad \forall v \in [n-1] \tag{6a}$$
$$\sum_{v \in [n]} A_{v,v} \geq n_0, \tag{6b}$$
$$2A_{u,v} \leq A_{u,u} + A_{v,v}, \qquad \forall u, v \in [n], \ u \neq v \tag{6c}$$
$$d_{v,v} = 0, \qquad \forall v \in [n] \tag{6d}$$
$$d_{u,v} \begin{cases} = 1, & A_{u,v} = 1 \\ > 1, & A_{u,v} = 0 \end{cases}, \qquad \forall u, v \in [n], \ u \neq v \tag{6e}$$
$$d_{u,v} \begin{cases} < n, & A_{u,u} = A_{v,v} = 1 \\ = n, & \min\{A_{u,u}, A_{v,v}\} = 0 \end{cases}, \qquad \forall u, v \in [n], \ u \neq v \tag{6f}$$
$$d_{u,v} \begin{cases} = d_{u,w} + d_{w,v}, & \delta_{u,v}^w = 1 \\ < d_{u,w} + d_{w,v}, & \delta_{u,v}^w = 0 \end{cases}, \qquad \forall u, v \in [n], \ u \neq v \tag{6g}$$
$$\delta_{v,v}^w = \begin{cases} 1, & w = v \\ 0, & w \neq v \end{cases}, \qquad \forall v \in [n] \tag{6h}$$
$$\delta_{u,v}^u = \delta_{u,v}^v = 1, \qquad \forall u, v \in [n], \ u \neq v \tag{6i}$$
$$\sum_{w \in [n]} \delta_{u,v}^w \begin{cases} = 2, & A_{u,v} = 1 \\ > 2, & A_{u,v} = 0, A_{u,u} = A_{v,v} = 1 \\ = 2, & \min\{A_{u,u}, A_{v,v}\} = 0 \end{cases}, \qquad \forall u, v \in [n], \ u \neq v \tag{6j}$$

Eq. (6) restricts $A_{u,v}, d_{u,v}, \delta_{u,v}^w$ in the following rules:

- Eq. (6a) forces nodes with smaller indexes exist.
- Eq. (6b) gives the lower bound of the number of existed nodes.
- Eq. (6c) means that there is no edge from node $u$ to $v$ if any of them does not exist.
- Eq. (6d) initializes the shortest distance from one node to itself, even it does not exist.

- Eq. (6e) forces the shortest distance from node $u$ and $v$ be $1$ if there is one edge from $u$ to $v$, and larger that $1$ otherwise.

  Rewrite Eq. (6e) as:

  $$d_{u,v} \leq 1 + n \cdot (1 - A_{u,v}), \quad \forall u, v \in [n], \ u \neq v$$
  $$d_{u,v} \geq 2 - A_{u,v}, \quad \forall u, v \in [n], \ u \neq v$$

  where $n$ is a big-M coefficient using the fact that $d_{u,v} \leq n$.

- Eq. (6f) sets the shortest distance from node $u$ to $v$ as $n$, i.e., $\infty$, if any of them does not exist. Otherwise, the shortest distance is less than $n$.

  Rewrite Eq. (6f) as:

  $$d_{u,v} \geq n \cdot (1 - A_{u,u}), \quad \forall u, v \in [n], \ u \neq v$$
  $$d_{u,v} \geq n \cdot (1 - A_{v,v}), \quad \forall u, v \in [n], \ u \neq v$$

- Eq. (6g) is the triangle inequality for the distance matrix $d$.

  Rewrite Eq. (6g) as:

  $$d_{u,v} \leq d_{u,w} + d_{w,v} - (1 - \delta_{u,v}^w), \quad \forall u, v, w \in [n]$$
  $$d_{u,v} \geq d_{u,w} + d_{w,v} - 2n \cdot (1 - \delta_{u,v}^w), \quad \forall u, v, w \in [n]$$

  where $2n$ is a big-M coefficient since $d_{u,w} + d_{w,v} \leq 2n$.

- Eq. (6h) initializes $\delta_{v,v}^w$ by definition, even node $v$ does not exist.

- Eq. (6i) initializes $\delta_{u,v}^u$ and $\delta_{u,v}^v$ by definition, even node $u$ or $v$ does not exist.

- Eq. (6j) makes sure that there is at least on node at the shortest path from node $u$ to $v$ if there is no edge from node $u$ and $v$ and these two nodes both exist. Otherwise, only $\delta_{u,v}^u$ and $\delta_{u,v}^v$ equal to $1$.

  Rewrite Eq. (6j) as:

  $$\sum_{w \in [n]} \delta_{u,v}^w \leq 2 + (n - 2) \cdot (1 - A_{u,v}), \quad \forall u, v \in [n], \ u \neq v$$

  $$\sum_{w \in [n]} \delta_{u,v}^w \leq 2 + (n - 2) \cdot A_{u,u}, \quad \forall u, v \in [n], \ u \neq v$$

  $$\sum_{w \in [n]} \delta_{u,v}^w \leq 2 + (n - 2) \cdot A_{v,v}, \quad \forall u, v \in [n], \ u \neq v$$

  $$\sum_{w \in [n]} \delta_{u,v}^w \geq A_{u,u} + A_{v,v} + (1 - A_{u,v}), \quad \forall u, v \in [n], \ u \neq v$$

  where $n - 2$ is a big-M coefficient since $\sum_{w \in [n]} \delta_{u,v}^w \leq n$.

To conclude, the formulation for shortest paths of all connected graphs with at least $n_0$ nodes and at most $n$ nodes is:

$$
\begin{cases}
\quad A_{v,v} \geq A_{v+1,v+1}, & \forall v \in [n-1] \\
\displaystyle\sum_{v \in [n]} A_{v,v} \geq n_0, & \\
\quad 2A_{u,v} \leq A_{u,u} + A_{v,v}, & \forall u,v \in [n],\ u \neq v \\
\quad d_{v,v} = 0, & \forall v \in [n] \\
\quad d_{u,v} \leq 1 + n \cdot (1 - A_{u,v}), & \forall u,v \in [n],\ u \neq v \\
\quad d_{u,v} \geq 2 - A_{u,v}, & \forall u,v \in [n],\ u \neq v \\
\quad d_{u,v} \geq n \cdot (1 - A_{u,u}), & \forall u,v \in [n],\ u \neq v \\
\quad d_{u,v} \geq n \cdot (1 - A_{v,v}), & \forall u,v \in [n],\ u \neq v \\
\quad d_{u,v} \leq d_{u,w} + d_{w,v} - (1 - \delta_{u,v}^w), & \forall u,v,w \in [n] \\
\quad d_{u,v} \geq d_{u,w} + d_{w,v} - 2n \cdot (1 - \delta_{u,v}^w), & \forall u,v,w \in [n] \\
\quad \delta_{v,v}^w = \begin{cases} 1, & w = v \\ 0, & w \neq v \end{cases}, & \forall v \in [n] \\
\quad \delta_{u,v}^u = \delta_{u,v}^v = 1, & \forall u,v \in [n],\ u \neq v \\
\displaystyle\sum_{w \in [n]} \delta_{u,v}^w \leq 2 + (n-2) \cdot (1 - A_{u,v}), & \forall u,v \in [n],\ u \neq v \\
\displaystyle\sum_{w \in [n]} \delta_{u,v}^w \leq 2 + (n-2) \cdot A_{u,u}, & \forall u,v \in [n],\ u \neq v \\
\displaystyle\sum_{w \in [n]} \delta_{u,v}^w \leq 2 + (n-2) \cdot A_{v,v}, & \forall u,v \in [n],\ u \neq v \\
\displaystyle\sum_{w \in [n]} \delta_{u,v}^w \geq A_{u,u} + A_{v,v} + (1 - A_{u,v}), & \forall u,v \in [n],\ u \neq v
\end{cases} \qquad \text{(MIP-SP-plus)}
$$

### A.4  PROOFS OF THEOREMS

***Proof of Theorem 3.5***.  If such $G$ exists, it is unique since $A_{u,v}$ gives the existence of every edge. Thus it suffices to show that $(d_{u,v}(G), \delta_{u,v}^w(G)) = (d_{u,v}, \delta_{u,v}^w)$ for $G$ defined with $A_{u,v}$.

We are going to prove it by induction on the shortest distance $sd$ from node $u$ to $v$ in graph $G$. Specifically, we want to show that for any $0 \leq sd < n$, and for any pair of $(u,v)$ such that $\min(d_{u,v}(G), d_{u,v}) = sd$, we have $d_{u,v}(G) = d_{u,v}$ and $\delta_{u,v}^w(G) = \delta_{u,v}^w,\ \forall w \in [n]$.

For $sd = 0$, $\min(d_{u,v}(G), d_{u,v}) = 0$ if and only if $u = v$. For any $v \in [n]$, it is obvious to have:

$$
\begin{aligned}
d_{v,v}(G) &= 0 = d_{v,v} \\
\delta_{v,v}^v(G) &= 1 = \delta_{v,v}^v \\
\delta_{v,v}^w(G) &= 0 = \delta_{v,v}^w,\ \forall w \neq v
\end{aligned}
$$

For $sd = 1$, consider every pair $(u,v)$ such that $d_{u,v}(G) = 1$, we have $A_{u,v} = A_{u,v}(G) = 1$, then it is easy to obtain:

$$
\begin{aligned}
d_{u,v}(G) &= 1 = d_{u,v} \\
\delta_{u,v}^w(G) &= 1 = \delta_{u,v}^w,\ \forall w \in \{u,v\} \\
\delta_{u,v}^w(G) &= 0 = \delta_{u,v}^w,\ \forall w \notin \{u,v\}
\end{aligned}
$$

where $\delta_{u,v}^w = 0,\ \forall w \notin \{u,v\}$ since:

$$
\sum_{w \notin \{u,v\}} \delta_{u,v}^w = \sum_{w \in [n]} \delta_{u,v}^w - \delta_{u,v}^u - \delta_{u,v}^v = 0.
$$

On the contrary, $d_{u,v} = 1$ gives $A_{u,v} = 1$, thus $A_{u,v}(G) = 1$ and $\delta_{u,v}^w(G) = \delta_{u,v}^w,\ \forall w$ by definition.

Now assume that for any pair of $(u, v)$ such that $\min(d_{u,v}(G), d_{u,v}) \leq sd$, we have $d_{u,v}(G) = d_{u,v}$ and $\delta_{u,v}^w(G) = \delta_{u,v}^w$, $\forall w$. Since $\delta_{u,v}^w(G) = \delta_{u,v}^w$, $\forall w \in \{u, v\}$ always holds by definition, we only consider $w \notin \{u, v\}$.

**Part 1**: We first consider every pair of $(u, v)$ such that $d_{u,v}(G) = sd + 1$. Since $sd + 1 \geq 2$, we know that $A_{u,v} = A_{u,v}(G) = 0$ and there exists $w \notin \{u, v\}$ on the shortest path from node $u$ to $v$ in graph $G$.

*Case 1.1*: For every $w \notin \{u, v\}$ such that $\delta_{u,v}^w(G) = 1$, since $d_{u,w}(G) \leq sd$ and $d_{w,v}(G) \leq sd$, we have:
$$d_{u,v} \leq d_{u,w} + d_{w,v} = d_{u,w}(G) + d_{w,v}(G) = d_{u,v}(G) = sd + 1.$$
The equality has to hold, otherwise, $d_{u,v} \leq sd$ gives $d_{u,v}(G) = d_{u,v} \leq sd$ by assumption. Therefore, $\delta_{u,v}^w = 1 = \delta_{u,v}^w(G)$.

*Case 1.2*: For every $w \notin \{u, v\}$ such that $\delta_{u,v}^w(G) = 0$, if $\delta_{u,v}^w = 1$, then $d_{u,w} + d_{w,v} = d_{u,v} = sd + 1$, which means that $d_{u,w} \leq sd$ and $d_{w,v} \leq sd$. By assumption, we have $d_{u,w}(G) = d_{u,w}, d_{w,v}(G) = d_{w,v}$ and then:
$$d_{u,w}(G) + d_{w,v}(G) = d_{u,w} + d_{w,v} = d_{u,v} = d_{u,v}(G).$$
which contradicts to $\delta_{u,v}^w(G) = 0$. Thus $\delta_{u,v}^w = 0$.

**Part 2**: Then we consider every pair of $(u, v)$ such that $d_{u,v} = sd + 1$. Similarly, we have $A_{u,v} = A_{u,v}(G) = 0$.

*Case 2.1*: For every $w \notin \{u, v\}$ such that $\delta_{u,v}^w = 1$, since $d_{u,w} \leq sd$ and $d_{w,v} \leq sd$, we have $d_{u,w}(G) = d_{u,v}$ and $d_{w,v}(G) = d_{w,v}$, then:
$$d_{u,v}(G) \leq d_{u,w}(G) + d_{w,v}(G) = d_{u,w} + d_{w,v} = d_{u,v} = sd + 1.$$
This equality also has to hold, otherwise, $d_{u,v}(G) \leq sd$, by assumption $d_{u,v} = d_{u,v}(G) \leq sd$, which is a contradiction.

*Case 2.2*: For every $w \notin \{u, v\}$ such that $\delta_{u,v}^w = 0$, if $\delta_{u,v}^w(G) = 1$, then $d_{u,w}(G) = d_{w,v}(G) = d_{u,v}(G) = sd + 1$, which means that $d_{u,w}(G) \leq sd$ and $d_{w,v}(G) \leq sd$. Therefore,
$$d_{u,w} + d_{w,v} = d_{u,w}(G) + d_{w,v}(G) = d_{u,v}(G) = d_{u,v},$$
which contradicts to $\delta_{u,v} = 0$. $\qquad\square$

***Proof of Theorem 3.6***. Fix the node number as $n_1$ with $n_0 \leq n_1 \leq n$, Eqs. (6a) – (6b) force:
$$A_{v,v} = \begin{cases} 1, & v \in [n_1] \\ 0, & v \in [n] \backslash [n_1] \end{cases}$$

substituting which to other constraints give us:
$$
\begin{aligned}
A_{u,v} = A_{v,u} = 0, & \qquad \forall u \in [n_1],\ v \in [n] \backslash [n_1],\ u \neq v \\
d_{v,v} = 0, & \qquad \forall v \in [n] \backslash [n_1] \\
d_{u,v} = d_{v,u} = n, & \qquad \forall u \in [n_1],\ v \in [n] \backslash [n_1],\ u \neq v \\
\delta_{u,v}^w = \delta_{v,u}^w = \begin{cases} 1, & w \in \{u, v\} \\ 0, & w \notin \{u, v\} \end{cases}, & \qquad \forall u \in [n_1],\ v \in [n] \backslash [n_1]
\end{aligned}
$$

One can easily check that all constraints associated with non-existed nodes are satisfied. Removing those constraints turns Eq. (MIP-SP-plus) into Eq. (MIP-SP) with size $n_1$. $\qquad\square$

## A.5 ENCODING FOR KERNEL OVER BINARY FEATURES

Assume that each graph $G$ has a binary feature matrix $F \in \{0, 1\}^{n(G) \times M}$, we need to formulate $k_F(F, F^i)$ and $k_F(F, F)$ properly. $k_F$ could be defined in multiple ways, here we propose a permutational-invariant kernel considering the pair-wise similarity among node features. Given two feature matrices $F^1, F^2$ corresponding to graphs $G^1, G^2$ resp., define $k_F$ as:
$$k_F(F^1, F^2) := \frac{1}{n_1 n_2 M} \sum_{v_1 \in V^1, v_2 \in V^2} F_{v_1}^1 \cdot F_{v_2}^2 = \frac{1}{n_1 n_2 M} \sum_{m \in [M]} N_m(F^1) \cdot N_m(F^2),$$

where $N_m(F^i) = \sum\limits_{v \in [n_i]} F^i_{v,m}$, $\forall m \in [M]$, and $n_1 n_2 M$ is the normalized coefficient.

Similar to Section 3.4, we have:

$$k_F(F, F^i) = \frac{1}{n n_i M} \sum_{m \in [M]} N_m(F^i) \cdot N_m,$$

where $N_m = \sum\limits_{v \in [n]} F_{v,m}$, $\forall m \in [M]$, and:

$$k_F(F, F) = \frac{1}{n^2 M} \sum_{m \in [M]} N_m^2 = \frac{1}{n^2 M} \sum_{m \in [M], c \in [M+1]} c^2 \cdot N_m^c,$$

where indicators $N_m^c = \mathbf{1}(N_m = c)$, $\forall m \in [M]$, $c \in [n+1]$ satisfy:

$$\sum_{c \in [n+1]} N_m^c = 1, \quad \sum_{c \in [n+1]} c \cdot N_m^c = N_m, \ \forall m \in [M].$$

### A.6 SIMPLIFY PATH ENCODING OVER UNDIRECTED GRAPHS

For undirected graphs, we first add the following constraints to guarantee symmetry:

$$\begin{cases} A_{u,v} = A_{v,u}, & \forall u, v \in [n], \ u < v \\ d_{u,v} = d_{v,u}, & \forall u, v \in [n], \ u < v \\ \delta_{u,v}^w = \delta_{v,u}^w, & \forall u, v, w \in [n], \ u < v \end{cases}$$

Since the inverse of any shortest path from node $u$ to $v$ is also a shortest path from node $v$ to $u$, for SSP and ESSP kernels, $D_s$, $\forall s \in [n]$ are even and we can fix odd indicators as zero:

$$D_s^c = \begin{cases} 1, & c \text{ is even} \\ 0, & c \text{ is odd} \end{cases}, \ \forall s \in [n], \ c \in [n^2 + 1].$$

Similarly, for SP and ESP kernels, we have:

$$P_{s,l_1,l_2} = P_{s,l_2,l_1}, \ \forall s \in [n], \ f_1, f_2 \in [L].$$

## B DISCUSSION

### B.1 CHOICE OF GRAPH KERNELS

Various graph kernels are proposed to better fit graph data. However, none of them could be incorporated as optimization constraints (nor are they designed for this purpose). Thus, current graph BO works mostly use evolutionary algorithms that generate candidates and then evaluate them, where graph kernels are used as graph functions to calculate the posterior mean and variance. The major difference here is that computing $k(G^1, G^2)$ given both $G^1$ and $G^2$ is quite easy, but representing $k(G^1, G^2)$ only given $G^1$ is super challenging since $G^2$ could be any arbitrary graph. BoGrape is built upon our theoretical contributions on encoding shortest paths into decision variables for arbitrary connected graphs. Therefore, it is not that we chose shortest-path kernel first for specific reasons then developed necessary formulations, but that the fundamental advances in graph optimization led us to shortest-path kernels.

### B.2 CHOICE OF ACQUISITION FUNCTIONS

BoGrape formulates acquisition optimization as a MIP, and LCB is chosen as a representative acquisition function given its popularity in BO and relatively simple form. BoGrape could be easily applied to other acquisitions functions in linear forms w.r.t. posterior mean and variance. For nonlinear acquisition functions, one could either use nonlinear solvers to optimize the resulting MINLP or linearize the acquisition functions. Since the acquisition only appears in objective, all graph-relevant constraints still work as before.

### B.3 EFFECTIVENESS OF ENCODING

Different shortest-path algorithms might affect the time complexity when computing the graph kernels, but they should be asymptotically similar since cost is dominated by computation of the shortest distance between any pair of nodes. For example, if we use Dijkstra's algorithm, which is a single-source shortest path algorithm, then we need to repeat it $n$ times and the complexity is $O(n(e + n \log n))$ with $e$ as the number of edges. Most importantly, the choice of shortest path algorithm is irrelevant to our shortest path encoding. Although our encoding is motivated by Floyd's algorithm, all constraints in our encoding are the necessary conditions that the shortest paths should satisfy no matter what algorithm is used. For the optimality of the encoding, our encoding builds a bijection between all connected graphs and all feasible solutions of Eq. (MIP-SP) as shown in Theorem 3.5, meaning it is optimal in terms of representations.

### B.4 COMPLEXITY ANALYSIS

The time complexity of computing all shortest paths for a graph with $n$ nodes is $O(n^3)$. When computing the covariance between two graphs (assume they both have $n$ nodes for simplicity), a naive implementation of shortest path kernel is $O(n^4)$, but our implementation is $O(nL^2)$ with $L$ being the number of node labels after storing $P_{s,l_1,l_2}(\cdot)$ as defined in Section 3.4. For other graph kernels, Random Walk (RW) (Gärtner et al., 2003) is $O(n^3)$, Subgraph Matching (Kriege & Mutzel, 2012) is $O(kn^{k+1})$ with $k$ as the size of subgraphs considered, and Weisfeiler-Lehman (WL) (Shervashidze et al., 2011) is $O(hm)$ with $h$ as the number of iterations and $m$ as the number of edges [7]. There are graph kernels with lower complexity than shortest-path kernels, but the complexity of calculating kernels in graph BO is less important than encoding the graph kernel as optimization constraints.

### B.5 LIMITATIONS

The major limitation of BoGrape (and most MIP-based methods) is the computational complexity. The BoGrape complexity stems from solving the MIP rather than computing kernel values. The tradeoff is: (i) BoGrape represents the whole, unavoidably large, search space precisely, and (ii) solving MIP to global optimality is time-consuming since proving optimality of a solution usually takes much more time than finding this solution. To better demonstrate this tradeoff, we perform an ablation study by varying the MIP time limit among $\{60, 600, 1200\}$ seconds on the molecular design case study on QM7 dataset with graph size $N = 10$. As Figure 5 illustrates, extending the computational time does not improve BO performance significantly. Nevertheless, Figure 6 shows that increasing time limits results in a smaller MIPgap, i.e. gives the the solver to more time prove a solution's optimality. In other words, finding good feasible solutions is easier (and important for practical BO performance), while closing the MIPgap (important for theoretical BO convergence) requires more computational effort.

### B.6 SCALABILITY OF BoGrape

Scalability issues always exist for combinatorial optimization, since the search space grows quickly. For BoGrape, there are several ways to improve scalability: (i) reduce the search space, e.g., only consider graphs that are similar to previous graphs in a trust-region fashion (similar in spirit to mutation over existing samples in evolutionary algorithms (Ru et al., 2021), or adversarial perturbations with limited budgets (Wan et al., 2021)), (ii) limit the solving time as we did in experiments, letting the MIP solver return the current best solution, (iii) develop computational heuristics for specific problems to identify promising candidates earlier, (iv) decompose large graphs into functional groups and optimize the graph structure over groups instead of nodes. Note that (iv) is frequently applied in graph tasks, e.g., cell-based neural architecture search (Wan et al., 2022), fragment-based molecular design (Zhang et al., 2024), etc..

### B.7 CHOICE OF APPLICATION

We choose the optimal molecular design task since (i) molecules can be represented as attributed, connected graphs, (ii) molecular properties, either measured or predicted, are suitable functions over graphs, and (iii) the MIP-based framework for molecular design is well-established. The baselines

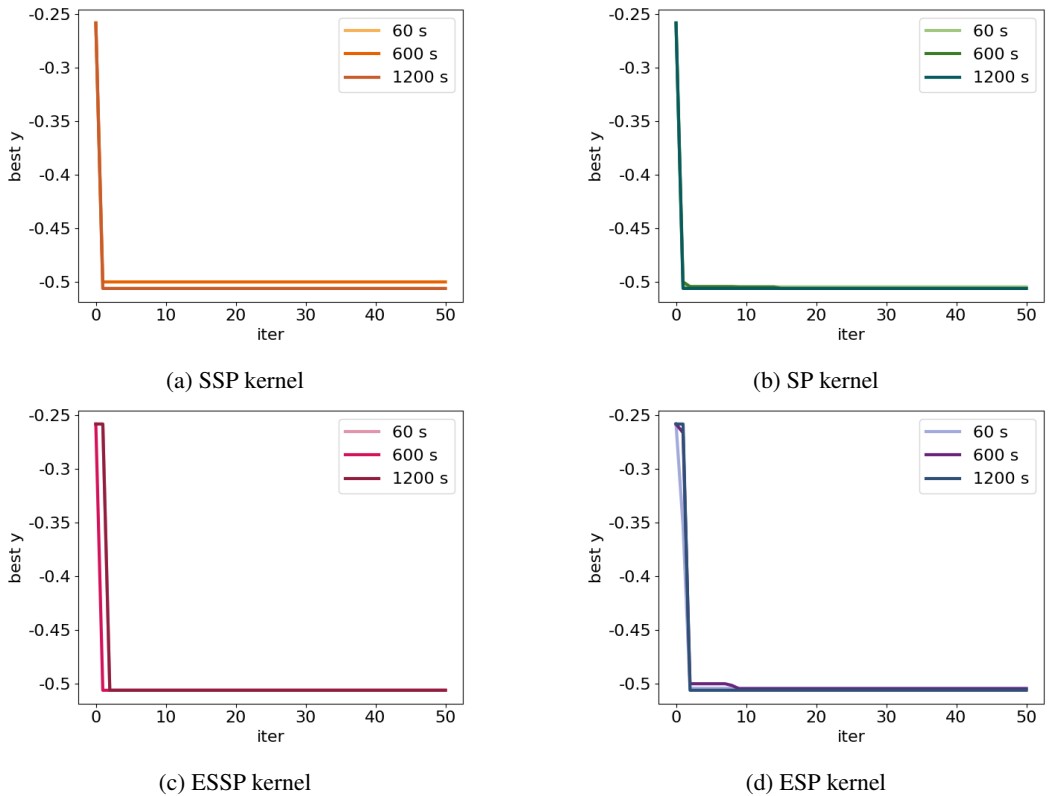

Figure 5: Performance of varying the time limit for BoGrape over QM7 datasets with graph size $N = 10$. Best objective is plotted at each iteration. Mean with 0.5 standard deviation over 10 replications is reported.

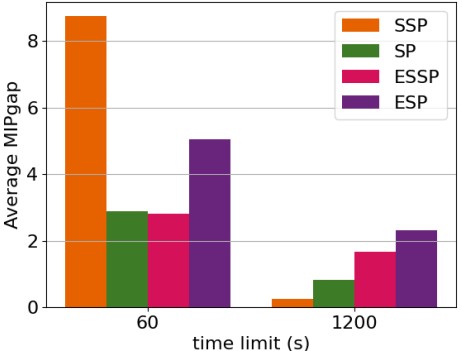

Figure 6: Comparison of the average MIPgap over all iterations when varying the time limit. Experiment conducted on QM7 dataset with graph size $N = 10$.

used in our experiments are less tailored to molecular design, and there are definitely more advanced methods. But the purpose of this case study is not showing BoGrape is a state-of-the-art method in molecular design, but investigating this problem from a constrained discrete optimization perspective. Meanwhile, although molecular design is a promising and important application area for BO (Paulson & Tsay, 2024), our proposed BoGrape procedure is general for any setting with functions defined over connected graphs.

## B.8 KERNEL SELECTION

Kernel selection is an interesting question explored in BoGrape. As discussed in Section 4.2, there is a trade-off between the kernel's expressiveness and the complexity of resulting optimization problems. With sufficient computational resources, more expressive kernels like ESP is preferred. But simpler kernels like SSP yield optimization problems that are easier to solve. As the graph size increases, linear kernels usually achieve better performance due to the overhead associated with formulating exponential kernels.

## C ADDITIONAL NUMERICAL RESULTS

### C.1 KERNEL PERFORMANCE

Besides four shortest-path graph kernels, we also test the performance of several classic graph kernels, including Random Walk (RW) (Gärtner et al., 2003), Subgraph Matching (SM) (Kriege & Mutzel, 2012), Weisfeiler-Lehman (WL) (Shervashidze et al., 2011), and Weisfeiler-Lehman Optimal Assignment (WLOA) (Kriege et al., 2016) kernels. To justify the effectiveness of the feature component in Eq. (3), we also test the combination of these four kernels with the same feature kernel used in shortest-path kernels. All GPs are trained by maximizing the log marginal likelihood. During GP training, we set bounds for kernel parameters, i.e., $\alpha, \beta, \sigma_k^2$, to $[0.01, 100]$ with 1 as their initial values, and set noise variance $\sigma_\epsilon^2$ as $10^{-6}$. $\beta_t^{1/2}$ defined in Eq. (1a) is set as 1.

Datasets QM7 (Blum & Reymond, 2009; Rupp et al., 2012) and QM9 (Ruddigkeit et al., 2012; Ramakrishnan et al., 2014) are used to test the kernel performance and train GNNs as graph functions. Each dataset consists of molecules with quantum mechanic properties. Following the setting in Zhang et al. (2024), we represent each molecule as a graph with $F = 15$ node features with $L = 4$ labels included, use the same structural constraints, and train a GNN as a predictor for each dataset. The trained GNN on QM7 has train and test errors of 0.0356 and 0.0337 respectively. Both the train and test errors of the GNN on QM9 are 0.0082. We provide an example of the node feature and label in such molecular graph to better distinguish the difference in their definitions:

*Example.* In the molecular design example on QM7 dataset, we followed the same setting as in Zhang et al. (2024). Each node (atom) has one label from $\{C, N, O, S\}$ and a feature vector with length $M = 15$, e.g. $(1, 0, 0, 0, 0, 1, 0, 0, 0, 0, 1, 0, 0, 1, 0)$ where the first four elements indicate the atom has label $C$, the $5^{th} - 8^{th}$ elements indicate the atom has two neighbors, the $9^{th} - 13^{th}$ elements indicate the atom is connected to 2 hydrogen atoms, the $14^{th}$ element indicates the atom is included in a double bond and the $15^{th}$ element indicates it is not included in a triple bond. More detailed definitions can be found in the Table 2 & 3 of Zhang et al. (2024).

Based on the molecular size $N$, we consider two settings (a) if the dataset includes molecules of size $N$, we randomly choose molecules from the dataset and use their real properties, and (b) for larger $N$, we use Limeade to generate molecules and use the trained GNN to predict their properties. To show the performance of different kernels on representing similarity between graphs, we apply setting (a) and perform a property prediction task using GPs equipped with the various kernels, shown in Figure 2. For larger graph sizes, we apply setting (b). The root mean square errors (RMSE) is reported in Figure 8 and Table 3, the mean negative log likelihood (MNLL) is reported in Table 4.

Two observations from these results are: (i) adding feature component largely improves the performance of all kernels in terms of predictive accuracy and uncertainty quantification, and (ii) our shortest-path kernels achieve comparable performance comparing to other graph kennels, which further justifies our choice of these kernels for global acquisition optimization.

## C.2 Ablation studies on the choice of $\beta_t$

BoGrape leverages the classical LCB acquisition function, where exploration and exploitation are balanced through its coefficient $\beta_t$ (Srinivas et al., 2010). We set $\beta_t = 1$ in our experiments for simplicity, as using constant values for $\beta_t$ is a standard approach in BO literature, e.g., Thebelt et al. (2021). Although changing $\beta_t$ does not affect the complexity of the acquisition optimization, we provide an ablation study on varying the value of $\beta_t$ as an investigation on the convergence behavior of BoGrape regarding different exploration-and-exploitation factor. We consider the same setup of the real-world case study on the QM7 dataset using SSP kernel with graph size $N = 10$ as in Section 4.3. We include three common choices of $\beta_t^{1/2}$ in BO literature: (1) 1 as in this work; (2) 1.96, e.g. Thebelt et al. (2021); (3) a time-dependent schedule of $3 \cdot \sqrt{0.5 \log(2(t+1))}$ as suggested by recent graph BO literature (Ru et al., 2021). We present the average minimal objective value (with 0.5 standard deviation in brackets) found over 50 iterations with 10 replications in Figure 9. This study confirms that, though there may indeed be value in tuning $\beta_t$ for a particular setting, BoGrape variants with different choices for $\beta_t$ largely exhibit similar convergence behavior. This observation justifies the choice on $\beta_t$ and further proves the robustness of BoGrape over key hyperparameters.

## C.3 Details for case study

Random sampling is a common baseline, but is excluded in Section 4.3 since it rarely produces even feasible solutions. Randomly sample feasible graphs is untrivial in molecular generation task because the graph structure and features should be reasonable and compatible with each other, e.g., satisfying structural feasibility, dataset-specific constraints, etc.. Here we consider random sampling over QM7 and QM9, to guarantee the feasibility of samples and compare it with Limeade (Zhang et al., 2025). Limeade is proposed as a feasible molecule generator, which is further enhanced by incorporating the composition constraints and symmetry-breaking constraints (Zhang et al., 2023). Figure 10 plots the regret curve over 50 iterations for both sample methods. In all cases, Limeade outperforms random sampling, showing the limitations of random sampling. Therefore, we choose Limeade as our sampling baseline.

For evolutionary algorithm, we apply the same random sampling and mutation procedure as in Ru et al. (2021). First, none of $10^6$ random samples is feasible, which is expected since the sample domain is $2^{N(N-1)/2} L^N$ ($\sim 10^{20}$ for $N = 10$, $\sim 10^{62}$ for $N = 20$), while the feasible domain is relatively much smaller. Then we give evolutionary algorithms $10^4$ feasible molecules generated by Limeade as an initial population and mutate each one 100 times, but only $0.35\%$ mutations are feasible for $N = 10$, and 3 out of $10^6$ are feasible for $N = 20$. Although a tailored evolutionary algorithm could be designed with better performance, it is neither the main focus of this work nor compatible with general settings. Therefore, we exclude WL-evol from baselines.

Given the trained GNNs used in Section C.1 as unknown graph functions, we conduct BoGrape to optimize them. In our experiments, 10 random molecules sampled by Limeade are used as the initial dataset, and 50 BO iterations are performed. We set PoolSearchMode=2 in Gurobi to generate feasible solutions using Limeade. For each BO run, we show the mean with 0.5 standard deviation of the best objective value over 10 replications. For the two baselines where we use Limeade as a sampling-based solver for the acquisition functions, we conduct an ablation study by increasing the number of candidates from 20 to 100. Figure 7 shows the results. While multiplying the number of candidates evaluated by five in each acquisition optimization step indeed improves the performance of the sampling-based baselines, BoGrape (which only proposes one sample per iteration) still outperforms Limeade. This ablation study emphasizes the importance of global acquisition function optimization. Besides the results shown in Figure 4, experimental results for $N \in \{15, 25\}$ are reported in Figure 11, supporting our analysis in Section 4.3.

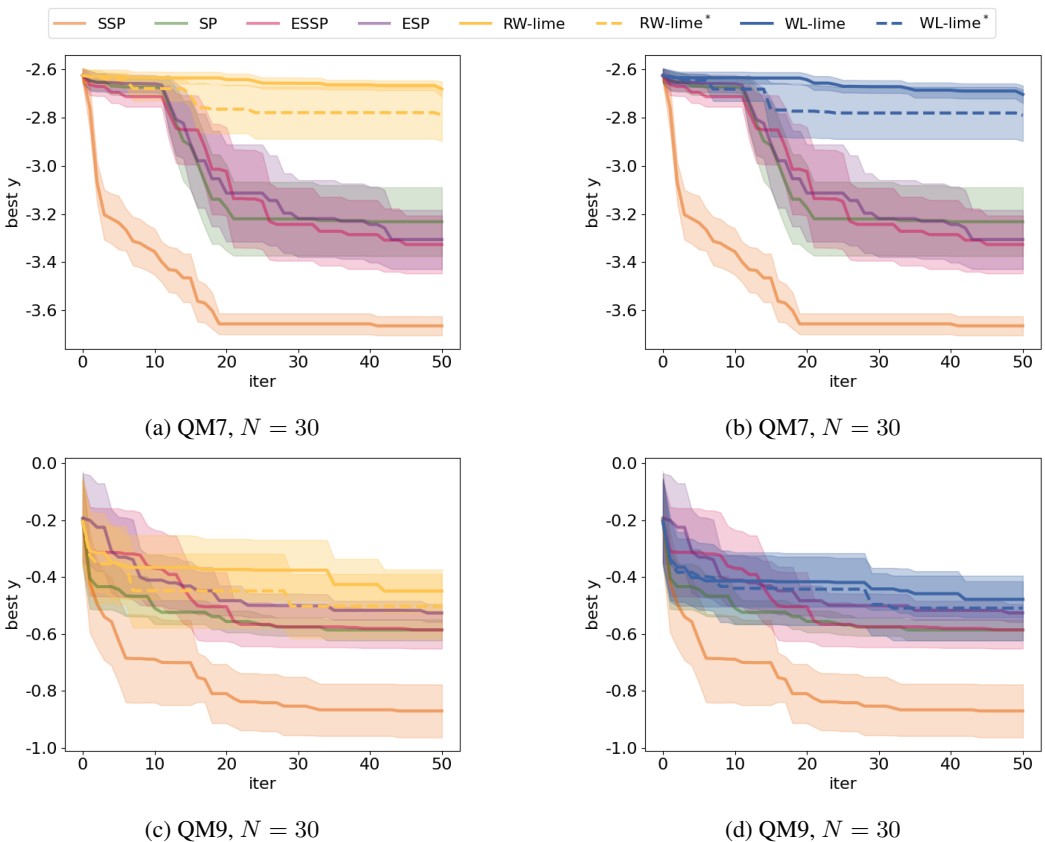

(a) QM7, $N = 30$

(b) QM7, $N = 30$

(c) QM9, $N = 30$

(d) QM9, $N = 30$

Figure 7: Performance of varying the number of samples used in acquisition optimization for baselines over QM7 and QM9 datasets with graph size $N = 30$. * indicates 100 candidates used in each iterations. Best objective is plotted at each iteration. Mean with 0.5 standard deviation over 10 replications is reported.

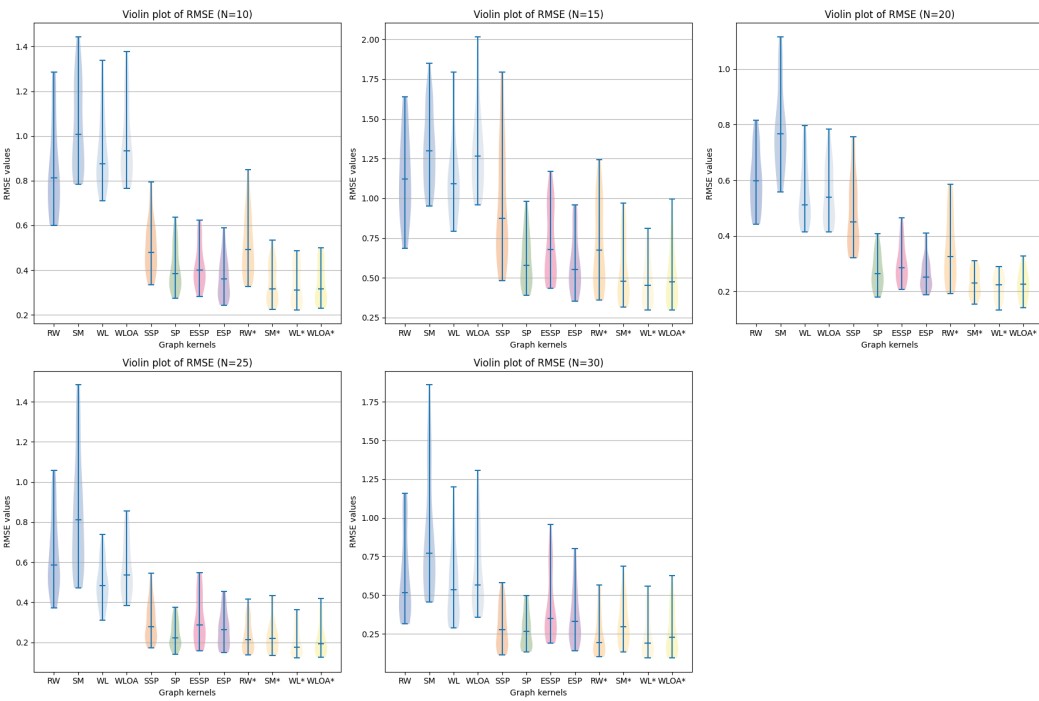

Figure 8: Violin plots to demonstrate GP regression performance on different graph kernels and graph sizes $N$. * indicates linear combination of given kernel and feature kernel. Each violin plots 25% percentile, median and 75% percentile of the RMSEs over the 100 replications.

Table 3: Model performance of GPs equipped with different graph kernels. * indicates linear combination of given kernel and feature kernel. For each graph size $N$, we use Limeade to random generate 20 and 100 molecules for training and testing, respectively, root mean square error (RMSE) of predictive error is reported over 100 replications.

| Kernel | N=10 | N=15 | N=20 | N=25 | N=30 |
|---|---|---|---|---|---|
| RW | 1.151(0.965) | 1.456(1.074) | 0.819(0.792) | 1.255(2.949) | 1.815(3.911) |
| SM | 1.301(0.945) | 1.693(1.221) | 1.398(4.284) | 1.678(3.041) | 2.816(6.667) |
| WL | 1.185(0.894) | 1.475(1.057) | 0.682(0.501) | 0.934(2.085) | 1.389(2.758) |
| WLOA | 1.256(0.930) | 1.612(1.084) | 0.682(0.508) | 1.002(2.075) | 1.492(2.750) |
| SSP | 0.710(0.680) | 1.810(2.819) | 0.682(0.683) | 0.638(1.701) | 1.122(3.085) |
| SP | 0.709(1.150) | 0.785(0.640) | 0.354(0.318) | 0.540(1.602) | 1.073(3.095) |
| ESSP | 0.619(0.772) | 0.928(0.732) | 0.436(0.550) | 0.766(1.615) | 1.862(4.907) |
| ESP | 0.539(0.628) | 0.790(0.691) | 0.341(0.304) | 0.757(1.980) | 1.574(4.516) |
| RW* | 0.826(0.922) | 0.984(0.920) | 0.520(0.613) | 0.523(1.723) | 0.995(2.655) |
| SM* | 0.519(0.670) | 0.765(0.699) | 0.290(0.301) | 0.593(1.487) | 1.278(3.748) |
| WL* | 0.498(0.636) | 0.723(0.731) | 0.278(0.286) | 0.562(1.676) | 1.006(2.646) |
| WLOA* | 0.545(0.710) | 0.866(1.002) | 0.289(0.293) | 0.608(1.702) | 1.049(2.648) |

Table 4: Model performance of GPs equipped with different graph kernels. * indicates linear combination of given kernel and feature kernel. For each graph size $N$, we use Limeade to random generate 20 and 100 molecules for training and testing, respectively, mean negative log likelihood (MNLL) is reported over 100 replications.

| Kernel | N=10 | N=15 | N=20 | N=25 | N=30 |
|---|---|---|---|---|---|
| RW | NA | NA | NA | NA | NA |
| SM | 15690.367(70415.316) | 793.739(3287.276) | 438.756(1797.462) | 3381.905(10099.946) | 36426.891(120815.092) |
| WL | 15688.598(70415.348) | 521.068(2243.726) | 304.246(1337.750) | 2944.284(9372.510) | 19793.178(69333.822) |
| WLOA | 15688.653(70415.362) | 521.248(2243.769) | 304.281(1337.685) | 2946.160(9372.893) | 19794.528(69333.027) |
| SSP | 15920.290(25866.469) | 7210.262(12025.188) | 1787.862(3453.177) | 1447.486(4738.234) | 13835.083(84912.369) |
| SP | 276.745(980.034) | 24.316(83.986) | 56.368(357.663) | 1090.370(4670.817) | 12862.266(84932.399) |
| ESSP | 53.093(154.109) | 530.289(3421.041) | 40.077(222.703) | 38.819(214.390) | 327.393(2336.968) |
| ESP | 4.962(17.396) | 1.877(3.650) | 0.658(1.308) | 2.597(7.894) | 248.222(2283.444) |
| RW* | NA | NA | NA | NA | NA |
| SM* | 2.863(9.559) | 2.974(9.078) | 0.845(2.916) | 9.611(31.583) | 1438.175(11278.539) |
| WL* | 2.572(9.418) | 2.764(9.118) | 0.477(2.357) | 3.782(15.643) | 702.551(6853.109) |
| WLOA* | 3.623(14.137) | 2.737(9.016) | 0.309(2.049) | 3.987(17.698) | 725.048(6974.228) |

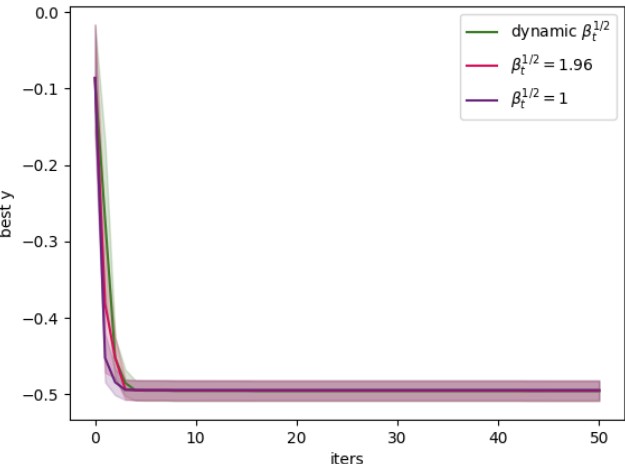

Figure 9: Bayesian optimization results on QM7 with $N = 10$ and different values of $\beta_t^{1/2}$. Best objective value is plotted at each iteration. Mean with $0.5$ standard deviation over 10 replications is reported.

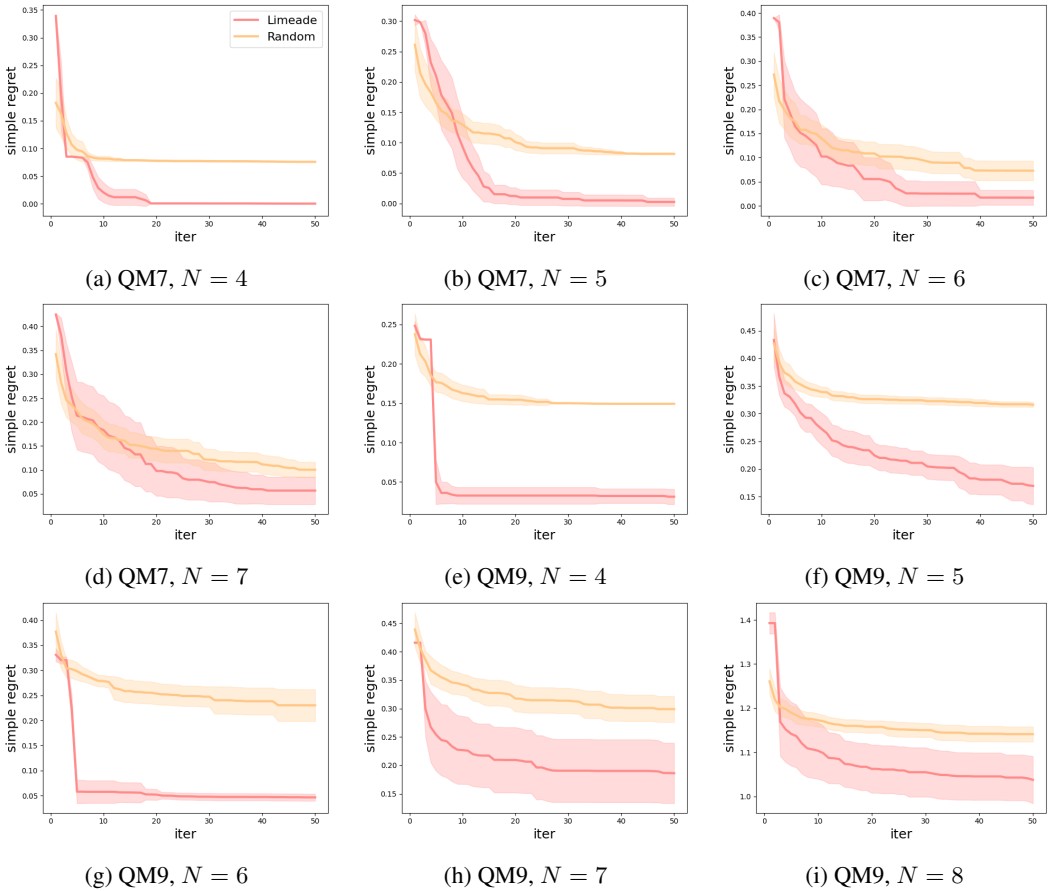

Figure 10: Performance of random sampling and Limeade over QM7 and QM9 datasets with different graph size $N$. Simple regret is plotted at each iteration. Mean with $0.5$ standard deviation over 10 replications is reported.

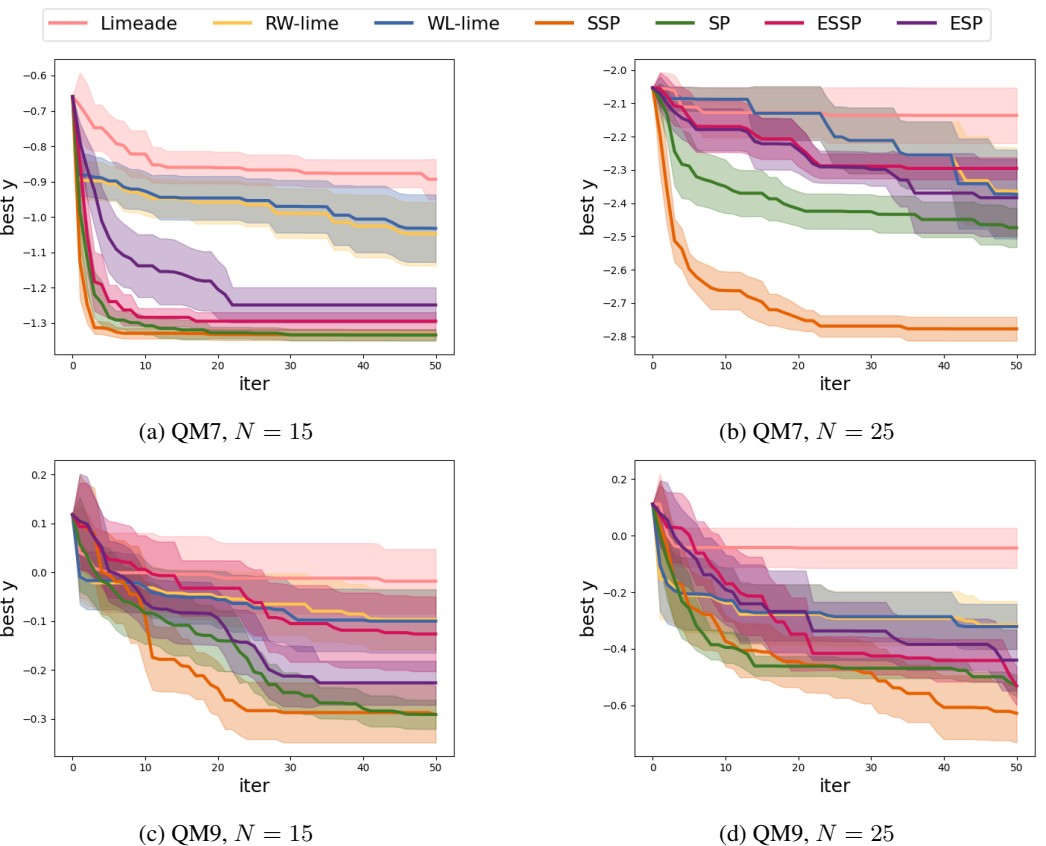

(a) QM7, $N = 15$

(b) QM7, $N = 25$

(c) QM9, $N = 15$

(d) QM9, $N = 25$

Figure 11: Bayesian optimization results on QM7 and QM9 with $N \in \{15, 25\}$. Best objective value is plotted at each iteration. Mean with $0.5$ standard deviation over 10 replications is reported.

