# OpenReview forum: "BoGrape: Bayesian optimization over graphs with shortest-path encoded"
_ICLR.cc/2026/Conference — ICLR 2026 Poster_

### Official Review · Reviewer_uSM7 · 2025-10-25

**Soundness:** 3
**Presentation:** 2
**Contribution:** 3
**Rating:** 6
**Confidence:** 3

**Summary:**

- This paper proposes a Bayesian optimization (BO) framework for optimizing black-box functions whose inputs are graphs, which commonly arise in domains such as molecular design, supply chains, and sensor placement.
- The authors study four variants of the shortest-path graph kernel (SP, SSP, ESP, and ESSP) and integrate them within a Gaussian process surrogate model.
- The main methodological contribution is the formulation of the acquisition function optimization as a mixed-integer programming (MIP) problem, which enables global optimization using off-the-shelf MIP solvers.
- The MIP has the acquisition function (in this paper, the authors consider the lower confidence bound, LCB) as its objective and includes constraints that ensure the predictive mean and variance from the GP model are consistent with the graph kernel, as well as additional constraints enforcing valid shortest-path relationships within the graph.
- The authors theoretically prove that each feasible MIP solution corresponds bijectively to a valid connected graph in the search space.
- The proposed framework, named BoGrape, is evaluated on both synthetic graph benchmarks and real-world molecular design tasks, demonstrating competitive or superior performance compared to baseline methods.

**Strengths:**

- This paper proposes a method to optimize an acquisition function in BO for a graph-input framework. This area of research is under-explored and has only a limited amount of literature.
- The method proposed here, i.e., formulating the acquisition function optimization problem as a mixed-integer programming (MIP) problem, is novel in the BO literature for graph functions.
- Formulating the acquisition function as an MIP also generalizes the capability of BO for graph functions, extending it from optimizing functions defined over graph nodes to functions defined over graph structures.
- The authors also provide a fruitful discussion on how to generalize the framework to other scenarios, such as graph structures with varying but bounded sizes and other nonlinear acquisition functions.

**Weaknesses:**

- The presentation of the paper needs improvement. Many typos and a wrong reference in the main part cause some confusion. For example, there is no Eq. (5) in the paper, and it seems that the authors were trying to refer to Eq. (MIP-SP).
- The main limitation of this paper is the high computational cost of solving the mixed-integer programming (MIP) formulation, which may restrict scalability to large graph sizes or complex kernels. More discussion on this limitation is required. This includes considering larger problems and presenting the comparison in a more informative way, for example by showing the Pareto front between acquisition optimization runtime and BO performance.
- There is a lack of explanation on the choices of parameters used in the competitor methods. For example, for methods that sample random graphs, why did the authors consider only 20 candidate graphs in each iteration? Since random sampling is cheap, it would be nice to see an ablation study with higher numbers.
- It is unclear how feasibility constraints of valid molecules are encoded in the method; more explanation on this part is needed.
- The authors considered several kernels but did not discuss the pros and cons of each of them. This discussion would help readers choose kernels that encode the properties relevant to their applications.

**Questions:**

- In the introduction, it is stated that there are two scenarios of graph optimization problems: optimizing over nodes and optimizing over graph structure, and this paper considers only the second case. If the algorithm is not applicable to the first case, this point should be specifically mentioned in the abstract to avoid overclaiming the ability of the framework. If the framework can be generalized to the first scenario, a brief idea on how to do it should be discussed.
- It is difficult to distinguish between node labels and node features. An explanation through an example would be helpful.
- The notation $\beta$ is used twice: in the objective of the MIP in Eq. (1a) and in the graph kernel weights in Eq. (3). I suggest changing the notation for clarity.
- In the last two constraints of (MIP-SP), what indices are the summations over? $w$?
- Should the $\alpha$ and $\beta$ in Eq. (3) be positive?
- Line 290: superscript $I$.
- Line 411: What is NAS? Network Architecture Search?

---

> ### Author Response · Authors · 2025-11-21
>
> Thanks for supporting BoGrape's methodological novelty and generalizability. To address the concerns:
>
> **[Equation references] (Weakness 1)** Apologies for the confusion, since many of our technical contributions are left in the Appendix due to the page limit. Eq.(5) indeed exists (Appendix A.2) and is not a wrong reference. Eq.(5) presents necessary conditions our graph variables should satisfy, and Eq.(MIP-SP) provides the corresponding encoding as MIP constraints. We are only able to present the final outcome, i.e. the MIP encoding Eq.(MIP-SP), given the limited space in the main paper, but we will revise the paper with clearer references to relevant sections in the Appendix to avoid this confusion.
>
> **[Scalability and computational cost] (Weakness 2)** Please refer to our general response to all Reviewers for the scalability of BoGrape and to Appendix B.5 for BoGrape limitations. The high computational cost of solving MIP is indeed a limitation, but it can be a fair trade-off for the guarantee of global optimality. In practice, the cost of evaluating the black-box function in BO is significantly more expensive than solving the acquisition function (e.g., consider molecular simulations or experiments). We would like to highlight BoGrape as a methodology- and theory-based paper on a *under-explored* field as the Reviewer summarized; nevertheless practical performance is indeed important. To study the tradeoff between acquisition optimization runtime and BO performance as suggested, we have added an ablation study on the MIP time limit in Appendix B.5, and reproduced below:
>
> To better demonstrate this tradeoff, we perform an ablation study by varying the MIP time limit among $\{60, 600, 1200\}$ seconds on the molecular design case study on QM7 dataset with graph size $N=10$. As Figure 5 illustrates, extending the computational time does not improve BO performance significantly. Nevertheless, Figure 6 shows that increasing time limits results in a smaller MIPgap, i.e., gives the the solver to more time prove a solution's optimality. In other words, finding good feasible solutions is easier (and important for practical BO performance), while closing the MIPgap (important for theoretical BO convergence)  requires more computational effort.
>
> **[Ablation study on the number of random sampling] (Weakness 3)** The number of $20$ samples is chosen as a heuristic parameter for the baselines, noting that BoGrape only generates one sample per iteration. It is an interesting idea to construct stronger baselines by increasing sample sizes. These strong baselines help further evaluate the performance of BoGrape, and we include this in Appendix C.3 of our revised version and also quote here:
>
> For the two baselines where we use Limeade as a sampling-based solver for the acquisition functions, we conduct an ablation study by increasing the number of candidates from $20$ to $100$. Figure 7 shows the results. While multiplying the number of candidates evaluated by five in each acquisition optimization step indeed improves the performance of the sampling-based baselines, BoGrape (which only proposes one sample per iteration) still outperforms them. This ablation study emphasizes the importance of global acquisition function optimization.

---

> ### Author Response · Authors · 2025-11-21
>
> **[Feasibility constraints] (Weakness 4)** Feasibility constraints of valid molecules are adapted from Limeade (Zhang et al., 2025), since it is also a MIP-based formulation. We introduced this encoding in Section 4.3 and further discuss it in Appendix C.3. To avoid this confusion, we will better document these feasibility constraints in a revised version.
>
> **[Choice of graph kernels] (Weakness 5)** Section 4.2 discusses how we choose different kernels for experiments. We provide a more extended discussion here: Kernel selection is an interesting question explored in our work. As discussed in Section 4.2, there is a trade-off between the kernel's expressiveness and the complexity of resulting acquisition optimization problems. With sufficient computational resources, we prefer more expressive kernels like ESP. But simpler kernels like SSP yield optimization problems that are easier to solve. As the graph size increases, linear kernels usually achieve better performance due to the overhead associated with formulating exponential kernels. We have also extended this discussion in the revised paper.
>
> **[Problem setting] (Question 1)** Thanks for the suggestion on clarifying our contribution. We do state the problem setting of BoGrape is ''to develop a principled framework for acquisition optimization over unseen graph structures'' in the abstract. BoGrape focuses on *optimizing over graph structure* as clarified in our introduction and does not generalize to the other case. We will rephrase the abstract to be more clear in a revised version.
>
> **[Definitions of node labels and node features] (Question 2)** We define these parameters in Appendix C.1, but we agree with the reviewer than an example would be useful and we will augment the paper with an example. Node labels indicate the type of node, and node features are application-dependent feature vectors associated with each node. Node labels are required for the SP and ESP graph kernels, and node features are used in feature kernel $k_F$ (see Appendix A.5). In practice, node labels are usually one-hot encoded as part of node features.
>
> *Example:* In the molecular design example on QM7 dataset, we followed the same setting as in Zhang et al., 2024. Each node (atom) has one label from $\{C, N, O, S\}$ and a feature vector with length $M=15$, e.g., $(1, 0, 0, 0, 0, 1, 0, 0, 0, 0, 1, 0, 0, 1, 0)$ where the first four elements indicate the atom has label $C$, the $5^{th}-8^{th}$ elements indicate the atom has two neighbors, the $9^{th}-13^{th}$ elements indicate the atom is connected to 2 hydrogen atoms, the $14^{th}$ element indicates the atom is included in a double bond, and the $15^{th}$ element indicates it is not included in a triple bond. More detailed definitions can be found in the Table 2 \& 3 of Zhang et al., 2024. As suggested, we have added this example in Appendix C.1 to help the readers better understand definitions.
>
> **[Notation of $\beta$] (Question 3)** The usage of $\beta_t$ is a standard definition in LCB. We can rename the two kernel parameters from $\alpha$ and $\beta$ to $\alpha_G$ and $\alpha_F$ for better clarity, but would prefer to change these after the rebuttal period to avoid creating confusions in ongoing discussion with the Reviewers.
>
> **[Summations indices and superscript] (Question 4 \& 6)** Thanks for identifying our typos! These have been fixed in the revised version. Both of the summations are over $w \in [n]$.
>
> **[Ranges of $\alpha$ and $\beta$] (Question 5)** Yes, Appendix C.1 defines the ranges for these parameters.
>
> **[NAS] (Question 7)** NAS is an abbreviation for neural architecture search. We have revised this into the full name.
>
> **[References]**
>
> Zhang, S., Campos, J. S., Feldmann, C., Sandfort F., Mathea, S., \& Misener R. Augmenting optimization-based molecular design with graph neural networks. Computers \& Chemical Engineering, 2024
>
> Zhang, S., Feldmann, C., Sandfort F., Mathea, S., Campos, J. S., \& Misener R. Limeade: Let integer molecular encoding aid.  Computers \& Chemical Engineering, 2025.

---

> ### Author Response · Authors · 2025-11-27
>
> Thank you again for your helpful suggestions regarding additional experiments. As a summary, based on your recommendations, we have added (in addition to the general responses above):
>
> - an investigation of the trade-off between acquisition optimization runtime and BO performance.
> - an ablation study on the batch size for random sampling (giving a stronger baseline).
>
> We’d be grateful for any feedback on whether these help address your concerns, if time permits.

---

### Official Review · Reviewer_Tm62 · 2025-10-29

**Soundness:** 3
**Presentation:** 3
**Contribution:** 3
**Rating:** 6
**Confidence:** 3

**Summary:**

This paper presents BoGrape, a new Bayesian optimization framework designed for black-box functions defined over graphs. The authors introduce four shortest-path-based graph kernels: SP, SSP, ESP, and ESSP; and then use them in BO with a global acquisition optimization scheme formulated as a mixed-integer optimization problem. Experiments are carried out on both synthetic and real-world scenarios, where the authors validate their proposed kernels and use them for BO on QM7 and QM9 datasets.

**Strengths:**

- This paper is well-written with clear logic and is very easy to follow.

- Unlike previous works on graph BO, which use evolutionary algorithms to explore the search space, this paper proposes a global optimization framework over the feasible graph space based on mixed Integer programming. This idea sounds very novel to me.

- The proposed graph kernels based on shortest-path look solid to me.

- I am happy to see that the authors compared the performance of their proposed kernels to previous graph kernels (but why are the results in the appendix?)

- I am also glad to see that the authors use baselines based on graph BO rather than simply comparing to random sampling.

- The authors also provide further details on the computational complexity, scalability, and limitations of their proposed framework, which help the audience better understand the pros and cons of this new method.

**Weaknesses:**

While the authors said their method can be easily applied to other common acquisition functions, they only consider LCB in their methodology and experiments. Is it possible to extend BoGrape to UCB (this one should be straightforward) or EI?

Another limitation, as mentioned by the authors in the appendix, is the scalability issue when the underlying graph is large. The maximum size of the graphs used in the experiments is 25, but in many real-world applications (e.g., social networks and infrastructure networks), the network size may go up to $10^5$. In these cases, I am unsure whether using a global optimization based on MIP is more efficient than adopting evolutionary algorithms.

But I don't think this is necessarily a "weakness" of the current work, since most of the graph-level optimization algorithms suffer from the scalability problem. Overall, I have a positive evaluation of this work.

**Questions:**

- On Figure 2, why is there no error bar on the first plot for the SSP kernel?

- Instead of comparing among your proposed kernels in Figure 2, I think it will be better to show the performance of other graph kernels in the main text (I notice that they are in Appendix C) by making each subplot a bit smaller. The reason is that the audience will be more interested in how your proposed kernels compare to previous graph kernels, rather than solely comparing among the new ones.

- A minor suggestion: it's better to briefly mention the testing accuracy of the GNNs that are used as ground-truth.

- I am a bit curious about the learned values of $\alpha$ and $\beta$ in your kernels, e.g., results in Section 4.1 Model Performance. (I think they should reflect the "importance" of graph structure and node feature in the underlying function?)

- Another minor suggestion: please consider using a larger font size for x and y labels, ticks on x and y axes, titles, and legends in Figures 2, 3, and 4.

---

> ### Author Response · Authors · 2025-11-21
>
> Sincre thanks for the positive evaluations on BoGrape and its soundness and contributions. To address the Reviewer's questions:
>
> **[Choice of acquisition functions] (Weakness 1)** Please see Appendix B.2, where we justify LCB and discuss the possibility of applying BoGrape with other acquisition functions. The acquisition function is the objective, so changing the objective does not affect the graph and the corresponding kernel encodings, which are constraints. BoGrape is not specific to LCB and only assumes that the acquisition function can be expressed analytically (in terms of the GP posterior mean and variance). Extending BoGrape to UCB is indeed straightforward, as one only needs to change the ''-'' sign in LCB to ''+'' and maximize rather than minimize. Applying BoGrape to EI is less straightforward, as defining EI involves a CDF term. However, we could develop a piecewise linear approximation of the CDF term and incorporate that into the objective function, using our proposed BoGrape formulation for the graph encoding constraints.
>
> **[Scalability] (Weakness 2)** Thanks for pointing out the scalability issue as a common challenge in graph optimization. We  refer the Reviewer to our general response to all Reviewers for further discussion on the BoGrape scalability. A minor point to clarify is that our experiments involve graph size up to $30$.
>
> **[Error bar of SSP kernel] (Question 1)** The error bars of SSP kernel are too small to be observed in this visualization because SSP kernel has relatively weak uncertainty quantification. This observation, which we discuss in Section 4.1, is further supported by Table 4, where SSP kernel has the largest MNLL. However, we agree that Figure 2 looks a bit strange, so we have now clarified in the updated Figure 2 caption the weak uncertainty quantification of the SSP kernel.
>
> **[Suggestions on paper presentation] (Question 2, 3 \& 5)** We appreciate these suggestions on moving more sections from our Appendix to the main paper. We admittedly had to move interesting results to the Appendices to keep the paper focused and fit the page limit. We have revised the paper with test accuracy of GNNs used (Appendix C.1) and enlarged the texts in Figures 2. We will  replace Figure 2 with a complete kernel comparison if space is available in a final version, and will also enlarge the text in Figure 3 and 4 as well.
>
> **[Learned kernel parameters] (Question 4)** Under the same setting as in Section 4.1, the learned values of $\alpha$, $\beta$ and signal variance $\sigma_k^2$ for exponential kernels are as follows:
>
> |      | $\alpha$ |  $\beta$ | $1/\sigma_k^2$ |
> |:----:|:--------:|:--------:|:--------------:|
> |  SSP |  99.9929 |  62.0424 |        -       |
> |  SP  |  92.4511 |  39.9165 |        -       |
> | ESSP |  2.29032 |  1.27791 |     14.933     |
> |  ESP | 0.966303 | 0.526128 |     35.237     |
>
> They indeed reveal the ''importance'' between graph topologies and graph features. In our case, the learned weight for graph kernels outweighs that of feature kernels in all kernel cases, further showing the importance of graph topologies. While our main focus is not the graph GP performance (but rather on optimization), are happy to add this observation to the paper if the Reviewer finds this result interesting and insightful.

---

> > ### Comment · Reviewer_Tm62 · 2025-11-21
> >
> > I'd like to thank the authors for their responses. I have no further questions and will keep my positive rating. Good luck!

---

### Official Review · Reviewer_bwgA · 2025-10-31

**Soundness:** 3
**Presentation:** 3
**Contribution:** 3
**Rating:** 6
**Confidence:** 5

**Summary:**

This paper proposes a graph-based Bayesian optimization that efficiently solves graph-level optimization problems using shortest-path encoding and mixed-integer programming (MIP). It establishes a bijective mapping between shortest-path MIP encoding and the set of connected graphs, thereby addressing the challenge of modeling graph structural constraints. Additionally, it introduces four variants of shortest-path kernels and a mixed kernel formulation to balance structural and feature similarity, and achieves constrained global acquisition optimization over mixed discrete–continuous spaces, with validated effectiveness on both synthetic and molecular design tasks.

**Strengths:**

1. This paper proposes a shortest-path-based MIP encoding strategy that establishes a strict bijection between connected graphs and the optimization constraint space.
2. It designs four positive-definite graph kernel variants and a mixed kernel formulation to adapt to different graph scales and nonlinear characteristics, integrating structural and attribute similarities.
3. It achieves global acquisition optimization within the MIP framework, explicitly modeling problem constraints and avoiding the local optima issues of traditional heuristic methods.

**Weaknesses:**

1. Since the computational complexity of MIP solving increases exponentially with the number of nodes, the current method only supports graphs with ≤30 nodes, making it challenging to handle large-scale graph scenarios.
2. The experiments mainly focus on undirected connected graphs, with insufficient validation of adaptability to directed and disconnected graphs, and a lack of related experiments, as well as detailed discussion and verification.

**Questions:**

1. The current MIP encoding only supports undirected connected graphs with no more than 30 nodes, and its scalability to large graphs has not been fully verified. Additionally, the proposed extensions for directed and disconnected graphs lack empirical support. It is recommended that the authors include corresponding validation experiments to enhance the method's generality and practicality.
2. The current experimental benchmarks do not include domain-specific Bayesian optimization methods, such as those for molecular design. The authors are encouraged to incorporate domain-relevant baselines to more comprehensively evaluate the performance and practical applicability of the proposed approach.

---

> ### Author Response · Authors · 2025-11-21
>
> Sincere thanks for the kind summary of the BoGrape contributions. To address the concerns:
>
> **[Scalability] (Weakness 1 \& Question 1)** Please refer to our general response to all Reviewers for detailed discussion on the scalability of BoGrape and its applicability in practical graph optimization scenarios.
>
> **[Adaptability to  disconnected graphs] (Weakness 2 \& Question 1)** We indeed do not cover disconnected graphs: see on Line 89 that BoGrape only applies to connected graphs. We require strong connectivity for the proposed graph encoding and for Theorems 3.5 \& 3.6. Weakly connected digraphs and disconnected graphs are out of BoGrape's scope and may be interesting directions for MIP formulations.
>
> **[Adaptability to directed graphs] (Weakness 2 \& Question 1)**
> Directed strongly connected graphs can be handled in the BoGrape framework, but it is true that our computational results focus on undirected graphs (this is because of Appendix A.6). Molecular design is a classic application for evaluating the performance of optimization methods on strongly connected graphs. However, we agree that anything claimed in the paper should be supported in the experiments. Therefore, we will add a statement in the introduction clarifying that, while we choose shortest path kernels for their ability to model both undirected and directed graphs, BoGrape only addresses undirected graphs because of the derived results we can use in Appendix A.6.
>
> **[Baselines] (Question 2)** Please refer to our general response to all Reviewers regarding domain-specific methods.

---

> > ### Author Response · Authors · 2025-11-27
> >
> > Thank you again for your helpful suggestions clarifying our contributions, scalability of graph BO, and baselines. During discussions, we added an investigation of the trade-off between acquisition optimization runtime and BO performance, and an ablation study on the batch size for random sampling (giving a stronger baseline).
> >
> > We’d be grateful for any feedback on whether our added clarifications and experiments help address your concerns, if time permits.

---

### Official Review · Reviewer_shL1 · 2025-10-31

**Soundness:** 2
**Presentation:** 3
**Contribution:** 3
**Rating:** 4
**Confidence:** 4

**Summary:**

This paper introduces BoGrape, a novel framework for Bayesian optimization (BO) over graph-structured domains. The key idea is to encode shortest-path relations in a mixed-integer programming (MIP) formulation, enabling global optimization of acquisition functions over combinatorial graph spaces while incorporating explicit structural constraints. The method is evaluated on synthetic benchmarks and molecular design tasks, showing superior performance to several baselines including random search, evolutionary algorithms, and existing graph BO methods.

**Strengths:**

1.The proposed approach is novel. The paper presents the first unified framework that formulates BO directly over graph structures. The use of MIP for global acquisition optimization is technically innovative and allows explicit incorporation of hard structural constraints, which is critical for applications such as molecular design.

2.The authors provide formal proofs ensuring the feasible domain of the formulation is equivalent to the graph space consisting of all connected graphs. This gives BoGrape a solid theoretical grounding compared to heuristic approaches.

3.The paper is generally well-written, with a clear motivation and logical flow from problem definition to methodology and results.

**Weaknesses:**

Overall, the main issues of the paper lie in two aspects.

Scalability concerns:
The proposed MIP formulation scales poorly with the number of nodes O(n3) variables due to shortest-path encoding. The largest experiments involve small graphs (n < 20), limiting its practical applicability. See questions 1 and 2.

Limited baselines:
Comparisons mainly include random, evolutionary, and one BO-based baseline.
Missing comparisons with recent neural graph optimization methods. see question 3.

**Questions:**

1. How would BoGrape perform on larger graphs (e.g., n>50)?

2. In line 52, the paper states that existing methods are “incapable of efficiently exploring the search domain.”However, based on later sections, the proposed method does not appear to demonstrate clear improvements in efficiency.
Could the authors provide quantitative comparisons of training and inference time costs for all methods across different tasks to substantiate this claim?

3. Why were recent neural graph optimization methods not included for a more comprehensive and up-to-date comparison? For instance, in the molecular experiments (Section 4.3), it is recommended that the authors extend the comparison beyond Bayesian optimization and evolutionary algorithms to include recent domain-specific neural graph optimization methods. Incorporating these approaches would provide a more comprehensive and up-to-date evaluation and better demonstrate the practical potential of the proposed method in real-world applications.

---

> ### Author Response · Authors · 2025-11-21
>
> Many thanks for supporting the novelty, technical innovation, and theoretical contribution of BoGrape. To address the concerns:
>
> **[Scalability] (Weakness 1 \& Question 1)** Please see our general response to all Reviewers for a discussion on the scalability concerns. Our experiments involve graphs up to size $30$. Global optimization over larger graphs (e.g., $N > 50$) is extremely challenging for all graph optimization methods, even outside of the BO setting, considering the cardinality of the labeled connected graph search space. Nevertheless, BoGrape can still be practical by loosening the optimality criteria or allowing longer computational time in these challenging settings, see for example Figures 5 and 6 in the revised PDF, which show that BoGrape does not need to solve the MIP to global optimality to get reasonable results. BoGrape also provides a foundational framework on which approximations/heuristics can be developed.
>
> **[Clarification on efficiency] (Question 2)** Thanks for pointing out this unclear phrasing. We intended ''efficiency'' here to refer to *the ability of exploring the whole feasible domain* rather than *the computational time spent in finding optimal solutions or training and inference time*. We reworded this phrase as ''effectively exploring the search domain''. Effectiveness in exploring the search space doesn't prove one optimization algorithm can find a better solution within a given time limit, but BoGrape effectively searches over the entire node-labeled connected graph space
>
> **[Baselines] (Weakness 2 \& Question 3)** Please refer to our general response to all Reviewers for a discussion on other baselines in molecular design experiments. We would also appreciate if the Reviewer can direct us to specific examples of *the recent domain-specific neural graph optimization methods* to help us better resolve relevant questions.

---

> > ### Comment · Reviewer_shL1 · 2025-11-27
> >
> > Thank you for the authors' response. I remain concerned about the scalability of the method, but the authors' reply and Reviewer Tm62's perspective on this issue also hold merit.
> >
> > Given that scalability challenges are common across most graph-level optimization algorithms at present, and the authors have clearly acknowledged this limitation in the paper, this limitation may not diminish the paper's overall contribution.  I will raise my score.

---

### Author Response · Authors · 2025-11-21
**Response to common comments**

Thanks to all Reviewers for providing insightful comments on BoGrape. Here we respond to the comments that multiple Reviewers mention. We provide responses to individual Reviewer comments separately.

**[Scalability]**

As Reviewer Tm62 states, scalability is indeed a common challenge in combinatorial optimization settings such as the field of graph optimization. Therefore, the referenced scalability issue is more a known characteristic of the problem setting, rather than a BoGrape weakness. Nevertheless, we agree that practical performance is important. Appendix B.6 discusses several possible avenues to improve BoGrape scalability and Section 4 performs experiments up to graph sizes of 30. Note that, given its emphasis on sample efficiency, even standard (non-graph) BO is often ''practically limited to optimizing 10–20 parameters'' (Moriconi et al., 2020), and popular high-dimensional BO implementations employ approximations such as trust regions (TuRBO; Eriksson et al., 2019) or low-dimensional subspaces (SAASBO; Eriksson \& Jankowiak, 2021).

In line with these observations, and in cases with relatively small graph sizes, BoGrape is computationally tractable and provides the solid theoretical guarantees accompanying global acquisition optimization. For larger problems where optimality is impractical, BoGrape can still return good feasible graph solutions by partially solving large-scale MIPs or by employing MIP heuristics. See in Figure 5 of our revised PDF version of the paper (caption beginning *Performance of varying the time limit ...*) that using BoGrape as a MIP heuristic is effective: we take the best feasible solution found by the solver in a short period of time and get good BO performance despite not having the time to prove global optimality of the optimization problem. Future work may consider approximation methods, such as the above, to extend the graph BO framework to ''high-dimensional'' BO.

BoGrape fills in the gap where a principled BO framework over graph spaces from a discrete optimization viewpoint is missing. Highlights of BoGrape include  equivalently encoding the graph space consisting of all connected graphs and global optimizing the graph acquisition functions. Despite the scalability issues characteristic of this setting, BoGrape represents the first approach to tackle global optimization over black-box graph functions even in small and moderate graphs cases. As such, BoGrape provides a foundation for developing more scalable graph optimizers (see Appendix B.6 for possible options).

**[Other baselines for molecular design experiments ]**

BoGrape is a general graph BO approach and we use molecular design as a representative case study. Our experiments highlight the efficiency of BoGrape across both synthetic benchmarks and real-world applications. We include relevant baselines in our experiments and ablation studies in Appendix C.3 to justify excluding certain methods.  Appendix B.7 further documents our choice of molecular design as a real-life case study.

There are indeed many interesting approaches specifically for molecular design, but a full computational comparison is challenging since different approaches consider different, nuanced problem settings. Our work views molecular design as a heavily-constrained, expensive-to-evaluate, black-box graph optimization problem. In other words, we consider the two generalizable concerns of feasibility and optimality, but remain otherwise agnostic to the problem setting. Specialized methods such as string kernels for molecules do not consider the same general setting, but rather employ domain-specific encodings, and often use tailored procedures to learn the feasible space. While adapting domain-specific methods to our general setting may indeed be possible with engineering efforts, we want to maintain the focus of this paper as a theoretical- and methodology-based study of general graph optimization, rather than a particular application. An interesting future extension would indeed be a molecular-design-tailored variant of BoGrape and comparison to domain-specific methods.

**[References]**

Eriksson, D., Pearce, M., Gardner, J., Turner, R. D., \& Poloczek, M. Scalable global optimization via local Bayesian optimization. NeurIPS, 2019.

Eriksson, D., \& Jankowiak, M. High-dimensional Bayesian optimization with sparse axis-aligned subspaces. UAI, 2021.

Moriconi, R., Deisenroth, M. P., \& Sesh Kumar, K. S. High-dimensional Bayesian optimization using low-dimensional feature spaces. Machine Learning, 2020.

---

### Author Response · Authors · 2025-12-02
**Discussion period summary and thanks**

We thank all the Reviewers for providing supportive feedback throughout the rebuttal period. We would like to provide the AC with a final summary of our work, BoGrape, and the Author-Reviewer discussions.

BoGrape introduces a graph Bayesian optimization-based method for optimizing black-box functions over graph structures. Specifically, we develop a theoretical encoding of the graph space containing all strongly connected graphs with/without node features, and we prove a bijection between the resulting decision space and graph search space. The strengths of BoGrape are generally recognized by the Reviewers, including its:
- novelty (e.g. *the first unified framework that formulates BO directly over graph structures* commented by Reviewer shL1, *novel in the BO literature for graph functions* commented by Reviewer uSM7),
- theoretical contribution (pointed out as *formal proofs* by Reviewer shL1 and *a strict bijection* by Reviewer bwgA),
- generality (e.g. *generalizes the capability of BO for graph functions* commented by Reviewer uSM7),
- significance as a global optimizer (pointed out by Reviewer bwgA and Tm62),
- presentation (e.g. *clear motivation and logical flow* by Reviewer shL1, *well-written with clear logic* by Reviewer Tm62).

The Reviewers’ common concerns focus on the scalability of BoGrape and the baseline choices for our real-world case study, which we have discussed in detail in our **Response to common comments**. We are very grateful to Reviewers Tm62 and shL1 for their engagement in the early stage of the discussion period, who reemphasized their positive rating and raised their score, respectively, as
the concerns on scalability and baselines are mitigated, at least partially. We also thank Reviewer uSM7 for their positive initial comments, and suggestions for our newly added (1) investigation of the trade-off between runtime and BO performance (Appendix B.5), and (2) an ablation study on the batch size for random sampling (giving a stronger baseline) (Appendix C.3), which we believe improve our study.

Lastly, we would like to thank all the Reviewers again for their overall positive evaluations, and the ACs for their time evaluating BoGrape.

---

### Meta-Review · Area_Chair_SXxR · 2025-12-08

**Summary:**

Acceptance recommendations were initially given by 3 of the 4 reviewers, and the 4th left a comment indicating that they would raise their score.  Hence, I can treated this as a unanimous decision for acceptance.  The reviewers gave positive comments on how this is a novel and interesting approach to graph-based BO problems by encoding shortest-path relations in an MILP.  They also commented positively on other aspects including the rigor and writing.

**Reviewer Concerns:**

There are no obvious remaining concerns sufficient to change the decision.

**Reviewer Scores:**

The likely final scores would be 6/6/6/6 (though an 8 might be possible), which means clear acceptance.

---

### Decision · Program_Chairs · 2026-01-26

Accept (Poster)